# Coherent deglacial changes in western Atlantic Ocean circulation

Hong Chin Ng [1], Laura F. Robinson[1], Jerry F. McManus[2], Kais J. Mohamed[3], Allison W. Jacobel [2], Ruza F. Ivanovic [4], Lauren J. Gregoire [4] & Tianyu Chen[1,5]

Abrupt climate changes in the past have been attributed to variations in Atlantic Meridional Overturning Circulation (AMOC) strength. However, the exact timing and magnitude of past AMOC shifts remain elusive, which continues to limit our understanding of the driving mechanisms of such climate variability. Here we show a consistent signal of the $^{231}Pa/^{230}Th$ proxy that reveals a spatially coherent picture of western Atlantic circulation changes over the last deglaciation, during abrupt millennial-scale climate transitions. At the onset of deglaciation, we observe an early slowdown of circulation in the western Atlantic from around 19 to 16.5 thousand years ago (ka), consistent with the timing of accelerated Eurasian ice melting. The subsequent weakened AMOC state persists for over a millennium (~16.5–15 ka), during which time there is substantial ice rafting from the Laurentide ice sheet. This timing indicates a role for melting ice in driving a two-step AMOC slowdown, with a positive feedback sustaining continued iceberg calving and climate change during Heinrich Stadial 1.

[1] School of Earth Sciences, University of Bristol, Bristol BS8 1RJ, UK. [2] Department of Earth and Environmental Sciences, Columbia University and the Lamont-Doherty Earth Observatory of Columbia University, Palisades, NY 10964, USA. [3] Department of Marine Geosciences, University of Vigo, 36310 Vigo, Spain. [4] School of Earth and Environment, University of Leeds, Leeds LS2 9JT, UK. [5] School of Earth Sciences and Engineering, Nanjing University, Nanjing 210046, China. Correspondence and requests for materials should be addressed to H.C.N. (email: hn9381@bristol.ac.uk)

nstrumental records provide clear evidence that long-term observations are required to establish the processes controlling AMOC variations[1]. The only way to examine AMOC processes over longer time periods is to use paleoceanographic evidence from times when there were clear changes in the Earth system[2,3]. Indeed, there have been considerable efforts to reconstruct AMOC strength during the deglacial period to examine its relationship with pronounced changes in climate[2], the atmospheric-oceanic carbon budget[3,4] and ice sheet volume[2,5] associated with millennial-scale climate events—Heinrich Stadial 1 (HS1, ~19–15 thousand years ago/ka), the Bølling-Allerød (BA, ~15–13 ka), and the Younger Dryas (YD, ~13–11 ka). However, the nature of the interplay between AMOC, climate and ice sheet systems during this period remains an outstanding scientific question, in large part because of the complications of interpreting circulation proxy records from individual locations that might not be representative of basin-wide AMOC strength.

Existing approaches for reconstructing past AMOC strength have included radiocarbon ($^{14}$C)[3], grain size[6], paleo-geostrophy[7] and $^{231}$Pa/$^{230}$Th[2]. The use of sedimentary $^{231}$Pa$_{ex,0}$/$^{230}$Th$_{ex,0}$ (the activity ratio of $^{231}$Pa to $^{230}$Th unsupported by the decay of U in the sediment and corrected for post-depositional decay, hereafter $^{231}$Pa/$^{230}$Th) as a geochemical proxy for ocean circulation rate is based on the difference in oceanic residence time between the two rapidly scavenged isotopes: 50–200 years for $^{231}$Pa and 10–40 years for $^{230}$Th[8]. Lateral transport, including advection and eddy diffusion, allows greater movement of $^{231}$Pa, which stays dissolved in seawater longer than $^{230}$Th, giving rise to deviations of $^{231}$Pa/$^{230}$Th from the production ratio (~0.093), which are subsequently preserved by particles deposited on the seafloor[8]. Observations of $^{230}$Th and $^{231}$Pa in the modern ocean suggest reversible exchange between the nuclides adsorbed onto sinking particulate matter and those dissolved in seawater[9], and the resulting $^{231}$Pa/$^{230}$Th is proposed to reflect zonal integration of signal across large distances (>1000 km)[10] and mainly ~1 km of the overlying water column when there is active water-mass advection[11]. Therefore, in principle, sedimentary $^{231}$Pa/$^{230}$Th may record a depth-integrated, large-scale lateral transport signature that reflects aspects of the overall circulation of the overlying water column[2]. At present, the active southward advection of North Atlantic Deep Water (NADW), the deep component of AMOC, causes greater export of seawater $^{231}$Pa than $^{230}$Th from the mid-latitude and low-latitude Atlantic[10], giving rise to sedimentary $^{231}$Pa/$^{230}$Th ratios lower than the production ratio (<0.093)[12]. Deglacial changes in AMOC strength have the potential to modify the rate of $^{231}$Pa export (relative to $^{230}$Th), a shift that should be recorded in the $^{231}$Pa/$^{230}$Th of Atlantic sediments. The proposed link between sedimentary $^{231}$Pa/$^{230}$Th and AMOC rate is supported by a number of ocean transport models[13,14], with recent modelling work indicating that this relationship is particularly strong in the deep western North Atlantic[14].

A number of temporally well-resolved sedimentary $^{231}$Pa/$^{230}$Th time-series point towards millennial-scale changes in AMOC strength during the most recent deglacial period[2,15–17]. However, this conclusion is not obviously supported by some other Atlantic $^{231}$Pa/$^{230}$Th records from the equatorial[18], marginal[19], eastern intermediate-depth[20–22], and southern sites[23]. Such discrepancies are most likely related to the geochemical behaviour of $^{231}$Pa and $^{230}$Th in the ocean. In particular, seawater Pa is more effectively scavenged in areas of high opal and particulate flux[8]. Previous down-core studies have shown strong positive correlations between $^{231}$Pa/$^{230}$Th and both opal and bulk sediment fluxes, providing evidence that scavenging by opal and other particles has the potential to significantly influence sedimentary $^{231}$Pa/$^{230}$Th at some

locations, including the tropical[18] and South Atlantic[23]. Recent observations[24] also reveal enhanced scavenging of both $^{231}$Pa and $^{230}$Th near margins (boundary scavenging), nepheloid regions, and the Mid-Atlantic Ridge (MAR), due to elevated particle fluxes, high concentrations of resuspended sediment, and Fe-rich hydrothermal fluxes respectively[24]. Therefore, it is important to establish the role of scavenging when interpreting sedimentary $^{231}$Pa/$^{230}$Th records in the context of circulation. In addition, the apparent timing of the observed changes in $^{231}$Pa/$^{230}$Th reported in previous work suggests that deglacial changes are not in phase across the Atlantic[23] – further challenging the interpretation of $^{231}$Pa/$^{230}$Th as an integrative signal of overturning circulation and hampering the use of this approach as a tool for testing climate models and hypotheses.

The aim of this study is to constrain a coherent picture of AMOC strength over the deglacial period by examining the observational and interpretive discrepancies between $^{231}$Pa/$^{230}$Th records. This approach requires an evaluation of the location of the records, sediment chronologies, and geochemical controls on sedimentary $^{231}$Pa/$^{230}$Th. We compiled all thirty-three of the available deglacial sedimentary $^{231}$Pa/$^{230}$Th time-series including four new records (Supplementary Figs. 1, 2) within the Atlantic Ocean on a common chronology. We then tested the correlation of the $^{231}$Pa/$^{230}$Th with both particle and opal fluxes (Supplementary Method, Supplementary Figs. 3–5), and found that nineteen records show no strong relationship to either of these two variables. Six of these records are from the eastern intermediate depths (Supplementary Figs. 1, 2). These sites are most likely to be subjected to the influence of shallow circulation, although a consensus is yet to emerge on the changes of these shallower water masses and their relationship to the overall AMOC during the last deglaciation[25]. Therefore, we focus on the remaining thirteen records (including four new records) from the deep basin (50° N–3° S, 2.7–4.6 km) and Brazil margin (2° S, 2.25 km) (Fig. 1, Table 1) —these parts of the Atlantic are found to be more responsive to changes in AMOC strength associated with the waxing and waning of NADW formation[14,26]. The sites of these thirteen records are located both near and far from continental margins, within potential nepheloid regions and at the MAR (Fig. 1). Assessment of these combined thirteen records therefore minimises potential signals associated with scavenging, and provides an overview of past AMOC changes. Coherent changes in the West and deep high-latitude North Atlantic sites represent the best-constrained picture of the timing of AMOC shift on the millennial timescale, and therefore its relationship with the timing of ice sheet and climate changes during the last glacial termination.

## Results

**Coherent signal in the West and deep high-latitude North.** Our new dataset reveals that sediment cores from the West and deep high-latitude North Atlantic exhibit remarkably consistent $^{231}$Pa/$^{230}$Th changes both in timing and amplitude over the last 25 thousand years (kyr) (Fig. 2a). The $^{231}$Pa/$^{230}$Th observed during the Last Glacial Maximum (LGM, ~22–19 ka) range from 0.059 to 0.083 and are higher overall than the Holocene (<10 ka) values, which range from 0.041 to 0.065. High $^{231}$Pa/$^{230}$Th values close to or above the production ratio (~0.093) are observed during HS1 (~19–15 ka). When data are available, these sites display a marked decrease in $^{231}$Pa/$^{230}$Th from HS1 to BA (~15–13 ka), followed by a distinct increase during the YD (~13–11 ka). The cores that contain data through the Holocene exhibit a gradual decrease in $^{231}$Pa/$^{230}$Th from around 11 to 8 ka (Fig. 2a).

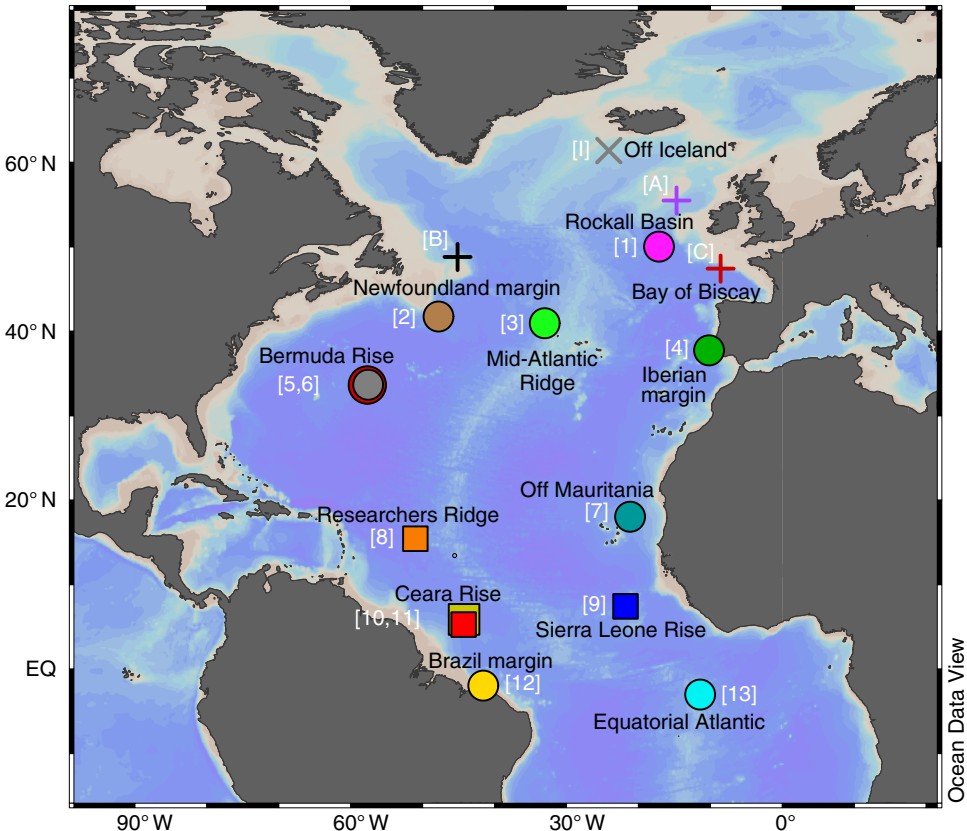

**Fig. 1** Location map of paleo-proxy records presented in the main discussion. These include sedimentary $^{231}Pa/^{230}Th$ records (square and circle symbols indicate new and previously published records respectively), sortable silt fraction (10–63 μm) grain size record (cross symbol), ice-rafted debris and meltwater discharge records ('plus' symbols). Numbers and letters in brackets denote the identity of the sediment records plotted in Figs. 2–4, with references listed in Table 1. The map was generated using the Ocean Data View programme[79]

**Table 1 Summary of sediment cores used to discuss deglacial changes in AMOC and ice sheet system**

| Site | Record | Core name | Latitude (° N) | Longitude (° E) | Water depth (km) | References | Notation on map/legend |
|---|---|---|---|---|---|---|---|
| Rockall Basin | $^{231}Pa/^{230}Th$ | SU90-44 | 50.02 | −17.10 | 4.279 | Gherardi et al.[16] | [1] |
| Newfoundland margin | $^{231}Pa/^{230}Th$ | MD95-2027 | 41.73 | −47.73 | 4.112 | Gherardi et al.[16] | [2] |
| Mid-Atlantic Ridge | $^{231}Pa/^{230}Th$ | IODP U1313 | 41.00 | −32.96 | 3.426 | Lippold et al.[23] | [3] |
| Iberian margin | $^{231}Pa/^{230}Th$ | SU81-18 | 37.77 | −10.18 | 3.135 | Gherardi et al.[15] | [4] |
| Bermuda Rise | $^{231}Pa/^{230}Th$ | OCE326-GGC5 | 33.70 | −57.58 | 4.550 | McManus et al.[2] | [5] |
| Bermuda Rise | $^{231}Pa/^{230}Th$ | ODP 1063 | 33.68 | −57.62 | 4.584 | Lippold et al.[77] | [6] |
| African margin–off Mauritania | $^{231}Pa/^{230}Th$ | MD03-2705 | 18.08 | −21.15 | 3.085 | Meckler et al.[78] | [7] |
| Researchers Ridge | $^{231}Pa/^{230}Th$ | JC094-GVY14 | 15.4643 | −50.9915 | 2.714 | This study | [8] |
| Sierra Leone Rise | $^{231}Pa/^{230}Th$ | JC094-GVY01 | 7.435 | −21.7963 | 3.426 | This study | [9] |
| Ceara Rise (northern) | $^{231}Pa/^{230}Th$ | EW9209-1JPC | 5.907 | −44.195 | 4.056 | This study | [10] |
| Ceara Rise (northern) | $^{231}Pa/^{230}Th$ | EW9209-3JPC | 5.313 | −44.26 | 3.288 | This study | [11] |
| Brazil margin | $^{231}Pa/^{230}Th$ | GeoB16202-2 | −1.9083 | −41.5917 | 2.248 | Mulitza et al.[17] | [12] |
| Equatorial Atlantic | $^{231}Pa/^{230}Th$ | RC24-12 | −3.01 | −11.417 | 3.486 | Bradtmiller et al.[18] | [13] |
| South of Iceland | Sortable silt | ODP 984 | 61.42 | −24.07 | 1.650 | Praetorius et al.[6] | [I] |
| Feni Drift, Rockall Basin | IRD | ODP 980 | 55.48 | −14.70 | 2.179 | McManus et al.[35]; Benway et al.[36] | [A] |
| Off Newfoundland | IRD | EW9302-2JPC | 48.7950 | −45.0848 | 1.251 | Marcott et al.[27] | [B] |
| Bay of Biscay | Fluvial discharge | MD95-2002 | 47.452 | −8.534 | 2.174 | Menot et al.[32] | [C] |

**Two-phase $^{231}Pa/^{230}Th$ transition during HS1**. An obvious feature of the newly combined West and deep high-latitude North Atlantic observations (Figs. 2a and 3b) is the pronounced increase in $^{231}Pa/^{230}Th$ towards or above the production ratio during HS1. Our compilation indicates that there may have been at least two phases in the observed HS1 transition. The first phase consists of a distinct early rise in $^{231}Pa/^{230}Th$ at the western sites from around 19 to 16.5 ka (Figs. 2a and 3b). In the second phase, widespread maximum values of $^{231}Pa/^{230}Th$ are observed in the West and deep North-East Atlantic from ~16.5 to 15 ka (Figs. 2a, b and 3b).

**LGM-deglacial contrast between the East and West Atlantic.** Our dataset also indicates contrasting trends in $^{231}Pa/^{230}Th$ between the East and West Atlantic over the last 25 kyr. All the western sites exhibit higher LGM $^{231}Pa/^{230}Th$ values than the late Holocene (<5 ka) values (Figs. 2a and 4). However, the eastern sites display different trends with water depths and latitudes. We observe

lower $^{231}Pa/^{230}Th$ at the Iberian margin (38° N, 3.14 km, Figs. 2b and 4) and off Mauritania (18° N, 3.09 km, Figs. 2b and 4), similar $^{231}Pa/^{230}Th$ at the MAR (41° N, 3.43 km, Figs. 2b and 4) and Sierra Leone Rise (7° N, 3.43 km, Figs. 2c and 4), and higher LGM $^{231}Pa/^{230}Th$ in the Equatorial Atlantic (3° S, 3.49 km, Figs. 2c and 4) and Rockall Basin (50° N, 4.28 km, Figs. 2a and 4). This inter-basin

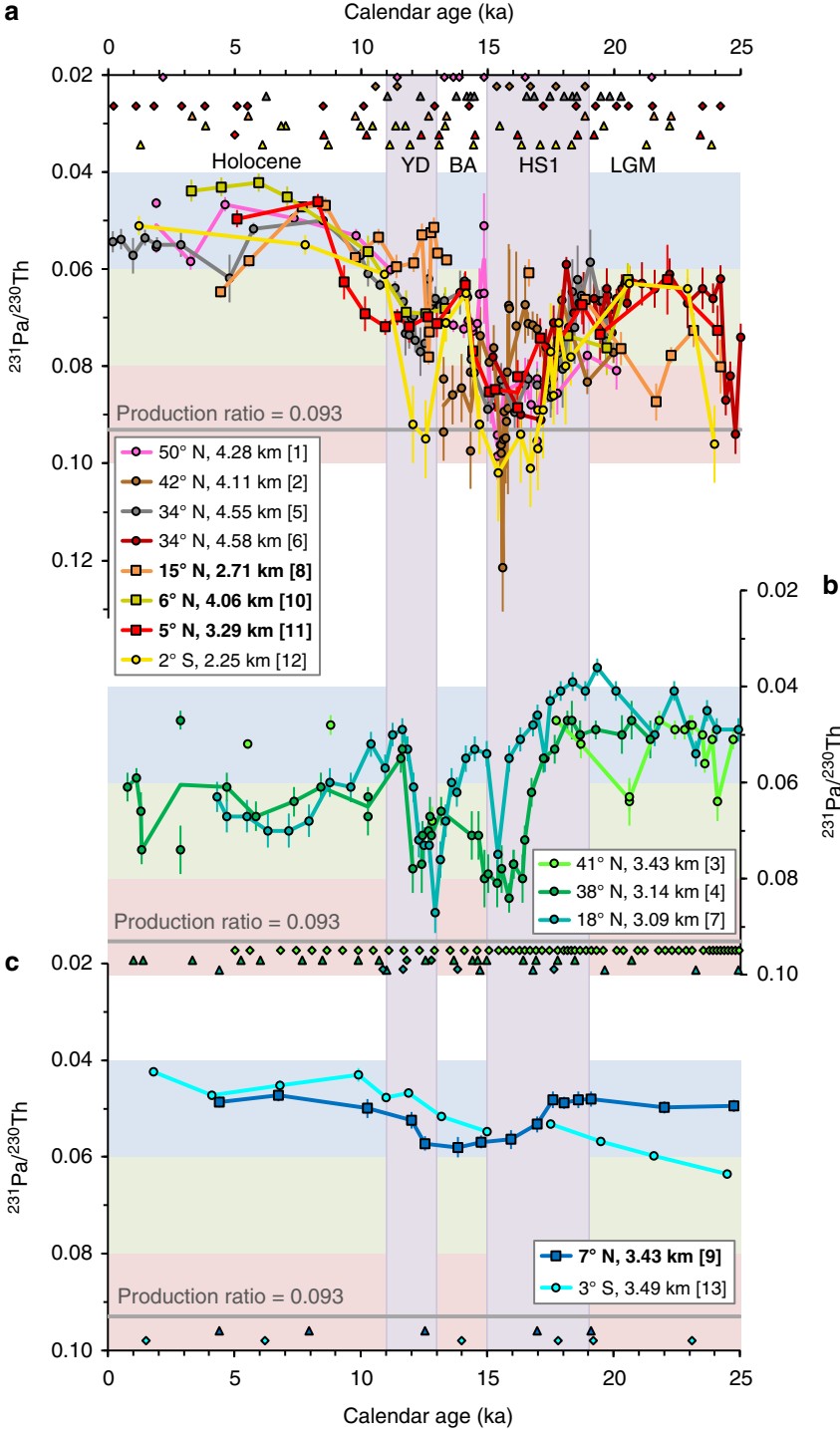

**Fig. 2** Atlantic sedimentary $^{231}Pa/^{230}Th$ records that reflect past changes in circulation rate. **a** West Atlantic and deep high-latitude (>50° N) North Atlantic records, **b** deep subtropical North-East Atlantic and Mid-Atlantic Ridge (MAR) records, and **c** deep low-latitude East Atlantic records. Error bars represent 2 s.e.m. Red, green and blue shading categorise the $^{231}Pa/^{230}Th$ ratios into high, middle and low values, respectively. Triangle and diamond symbols respectively signify $^{14}C$ and non-$^{14}C$ chronological tie-points. Bracketed numbers denote the identity of the sediment cores marked in Fig. 1, with references listed in Table 1. Bold characters in the figure legend and the square symbols indicate new $^{231}Pa/^{230}Th$ reconstructions from this study. Annotations of key climate events: LGM Last Glacial Maximum, HS1 Heinrich Stadial 1 (purple shading), BA Bølling-Allerød, YD Younger Dryas (purple shading)

contrast continued over the ensuing deglaciation at the low latitudes, with persistently low $^{231}$Pa/$^{230}$Th observed at Sierra Leone Rise (7° N, 3.43 km, Fig. 2c), even during HS1 when there was a substantial $^{231}$Pa/$^{230}$Th rise in the western basin (Fig. 2a).

## Discussion

Our new integrated $^{231}$Pa/$^{230}$Th dataset (Fig. 2) reveals large-scale patterns of AMOC change that are not dependent on any single-

core location and includes cores from diverse oceanographic and sedimentary settings (Fig. 1). Most importantly, the coherent $^{231}$Pa/$^{230}$Th signal in the West and deep high-latitude North Atlantic (Fig. 2a, Supplementary Data 3) provides robust evidence for millennial-scale changes in deep Atlantic circulation strength during the deglacial period. The temporal consistency of the integrated records differs from some prior observations, which found offsets in the timing of the deglacial $^{231}$Pa/$^{230}$Th shift in the

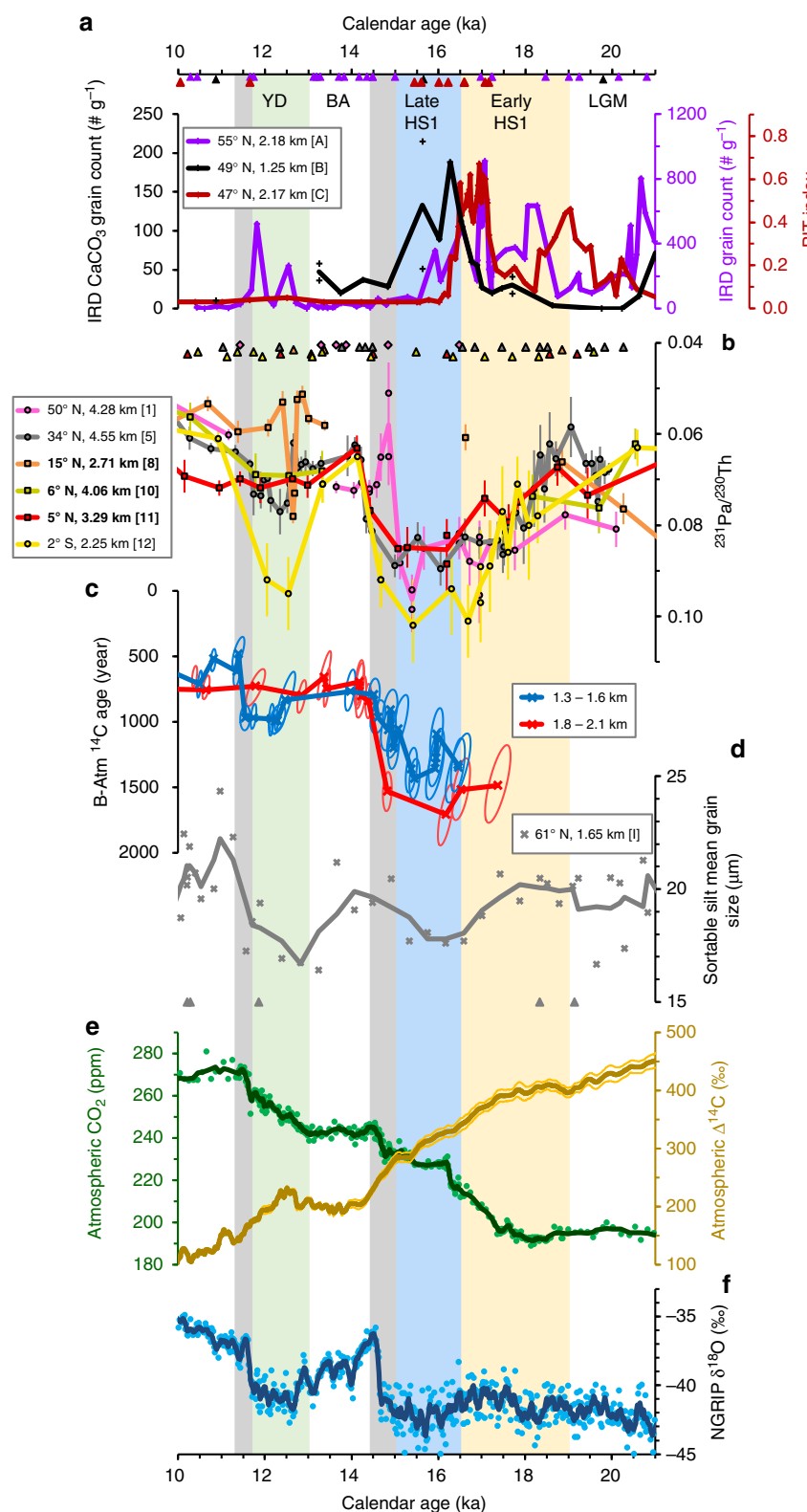

deep West Atlantic[23]. In particular, the widespread $^{231}$Pa/$^{230}$Th increase from the LGM to HS1 (Fig. 2a) indicates that there was an extensive, basin-scale control reducing $^{231}$Pa export during HS1. Our dataset strengthens the proposal[2] that this control was a reduction in deep-water transport in the North Atlantic.

One working hypothesis for millennial-scale Heinrich stadials such as HS1 relies on significant freshwater addition to the North Atlantic[27,28]. For example, meltwater associated with large ice-rafting events might alter surface water buoyancy leading to reduced deep-water production and AMOC slowdown. In turn, this would result in less northward heat transport, which would act to cool the Northern Hemisphere[28]. This hypothesis has sparked a long-standing debate over the mechanisms linking the ocean, climate and ice-sheet systems during Heinrich stadials. For example, a recent study from the North Atlantic emphasised that climate cooling began prior to the arrival of ice-rafted debris (IRD) layers, indicating large ice-rafting events may be a consequence of, rather than a trigger for Heinrich stadials[29]. This late arrival of the IRD layers is supported by other well-dated records from the North Atlantic[27,30]. In contrast, there is evidence for early meltwater discharge[31,32] and iceberg rafting from the Eurasian ice sheets[33,34] that may have initiated an AMOC slowdown leading to the development of HS1. Clearly, the timing and duration of AMOC slowdown is key to establishing potential triggers and forcing mechanisms linking such climatic events with ocean behaviour.

To examine the phasing of the ocean circulation and ice sheet systems during HS1, we compare our $^{231}$Pa/$^{230}$Th compilation (Fig. 3b) with Eurasian fluvial discharge and IRD records from the North Atlantic (Fig. 3a). Fluvial discharge from the Bay of Biscay (47° N)[32] shows peaks of meltwater from the Eurasian glaciers and ice sheets from ~19 to 16.5 ka. During this same period, the eastern IRD record from Feni Drift (55° N)[35,36], which is expected to be sensitive to ice rafting from the Eurasian ice sheets, displays peaks of IRD grain counts. In contrast, the western record off Newfoundland (49° N)[27] shows pronounced IRD CaCO$_3$ grain counts, indicative of ice rafting from the Laurentide ice sheet, from ~16.5 to 15 ka (Fig. 3a). These combined observations indicate that episodes of Eurasian meltwater discharge and iceberg calving occurred during early HS1 (~19–16.5 ka), whereas Laurentide iceberg calving is evident in late HS1 (~16.5–15 ka). The observed $^{231}$Pa/$^{230}$Th increase beginning ~19 ka (Fig. 3b, Supplementary Figs. 6, 7) is aligned with enhanced fluvial draining of the Eurasian ice sheets (Fig. 3a), supporting an early AMOC decline that was linked to Eurasian-sourced freshwater forcing[37,38]. Weakening of AMOC could cause subsurface warming of the Nordic seas that could further destabilise the Eurasian ice sheets[39], leading to increased iceberg calving activity as recorded in the Feni Drift core (Fig. 3a). Melting of these Eurasian icebergs potentially (Fig. 3a) contribute to progressive weakening of AMOC during early HS1 (Fig. 3b). This first phase of AMOC reduction (~19–16.5 ka) might eventually lead to more widespread subsurface warming that could reach the Labrador Sea[27], triggering ice-rafting events from the Laurentide ice sheet as recorded in the Newfoundland core (Fig. 3a). The timing of

this Laurentide iceberg flux coincides with widespread maximum $^{231}$Pa/$^{230}$Th values observed in the West and deep North-East Atlantic (Figs. 2a, b and 3b, Supplementary Figs. 6, 7), indicating that Laurentide iceberg calving might have sustained the second phase of maximum AMOC reduction during late HS1 (~16.5–15 ka). The phasing of AMOC, meltwater discharge and iceberg flux observed here suggests a two-phase mechanism for AMOC reduction at the beginning of the deglacial. The two-phase AMOC reduction has important implications for the HS1 climate, as it broadly coincides in timing with the two-phase changes in global hydrological cycle observed during this period[40].

The widespread decrease in $^{231}$Pa/$^{230}$Th in the West and deep North-East Atlantic sediment cores after peak values at around 15 ka (Figs. 2a, b and 3b, Supplementary Figs. 6, 7) signifies a recovery in the strength of deep-water transport at the onset of the BA. The inferred resumption in deep circulation and deep-water formation in the North Atlantic coincides in timing with abrupt Greenland warming (Fig. 3f) and major $^{14}$C ventilation of the Atlantic Ocean[3,41] (e.g. Fig. 3c). These proxy records are consistent with recent modelling studies[42–45], which confirm that AMOC is a key player in abrupt changes in climate and the ocean-atmospheric carbon budget[43]. The AMOC reinvigoration during the BA (~15–13 ka), evident from our $^{231}$Pa/$^{230}$Th dataset (Fig. 3b) and $^{14}$C record[3] (Fig. 3c), coincides with an interval of diminished IRD and Eurasian meltwater discharge (Fig. 3a). However, sea level records[46] and ice sheet modelling[47] suggest a considerable amount of ice melt from the Laurentide ice sheet entering the Atlantic Ocean during this period. If a significant amount of this freshwater flux reached the deep-water formation sites in the North Atlantic, this would be inconsistent with the hypothesis that AMOC recovery during the BA was due to reduced freshwater forcing from the northern ice sheets. In fact, rather than a driving factor of circulation change, the major Laurentide ice sheet melting and sea level rise might have been a consequence of reinvigorated AMOC during this interval, as previously inferred[2,47]. Alternatively, a recent coupled atmosphere-ocean model[4] suggests that the overall increase of atmospheric CO$_2$ from HS1 leading up to BA (Fig. 3e) could be the cause of AMOC reinvigoration, even in the event of increased freshwater forcing. The model shows that the atmospheric CO$_2$ increase could alter the hydrological cycle, leading to northward transport of more saline surface current in favour of deep-water formation in the North Atlantic[4].

A subsequent $^{231}$Pa/$^{230}$Th increase is observed during the YD, but overall it is of a smaller magnitude and duration compared to HS1 (Fig. 3b, Supplementary Figs. 6, 7). This difference suggests a reduction in Atlantic deep-water transport during the YD that was less intense or shorter than the reduction during HS1[2], or that sedimentation rates in some of these cores are not sufficient to prevent smoothing of the YD signal[2,16] (Supplementary Fig. 11). Notably, The YD peak of $^{231}$Pa/$^{230}$Th observed at the Researchers Ridge (15° N, 2.71 km) appears to be short-lived relative to the other sites (Fig. 2), potentially implying some spatial differences in the timing and duration of the reduction in water-mass transport

**Fig. 3** Paleo-proxy records of ice sheet, Atlantic circulation and climate changes from 21 to 10 ka. **a** North Atlantic ice-rafted debris[27,35,36] records and a proxy (terrestrial organic matter isoprenoid tetraether (BIT) index) record of Eurasian fluvial discharge[32], **b** West and high-latitude North Atlantic $^{231}$Pa/$^{230}$Th records that contain data over the Holocene, deglacial and LGM (see also Fig. 2a), **c** northern tropical Atlantic (5–15° N) coral proxy records of ocean radiocarbon (B-Atm $^{14}$C ages with ellipsoid error bars)[3], **d** sortable silt mean grain size record of Iceland-Scotland Overflow Water strength (an important component of NADW)[6], **e** atmospheric CO$_2$ record from the West Antarctic Ice Sheet Divide ice core (WDC)[51] and atmospheric Δ$^{14}$C record from the IntCal13 compilation[50], and **f** Northern Greenland ice core temperature proxy (δ$^{18}$O) record[55]. Triangle and diamond symbols respectively signify $^{14}$C and non-$^{14}$C chronological tie-points. Numbers and letters in brackets denote the identity of the sediment cores marked in Fig. 1, with references listed in Table 1. Bold characters in the figure legend and the square symbols indicate new $^{231}$Pa/$^{230}$Th reconstructions from this study. Yellow shading—early HS1, blue shading—late HS1, green shading—YD, grey shading mark the HS1-BA transition and the YD-early Holocene transition

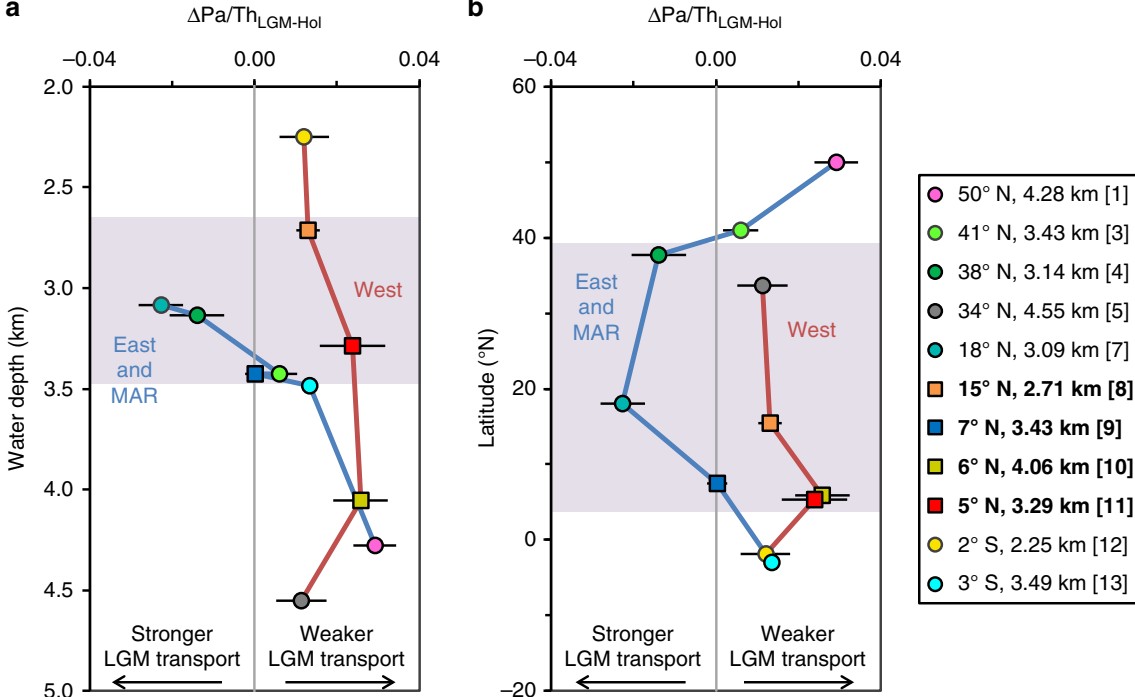

**Fig. 4** Differences in averaged $^{231}Pa/^{230}Th$ values between the LGM (~22–19 ka) and late Holocene (≤5 ka) ($\Delta Pa/Th_{LGM-Hol}$). **a** $\Delta Pa/Th_{LGM-Hol}$ versus water depth and **b** $\Delta Pa/Th_{LGM-Hol}$ versus latitude. The calculation was carried out based on the thirteen Atlantic $^{231}Pa/^{230}Th$ time-series selected for interpretation of deep Atlantic circulation (Fig. 2). The $\Delta Pa/Th_{LGM-Hol}$ plots have eleven data points as one Bermuda Rise core (34° N, 4.58 km) and the Newfoundland margin core (42° N, 4.11 km) do not have Holocene data. Purple shading indicates an East-West contrast in the $\Delta Pa/Th_{LGM-Hol}$ at mid-latitude and low latitudes of the North Atlantic. Error bars represent 95% confidence interval of the averaged $^{231}Pa/^{230}Th$ values, and represent 2 s.e.m when there is only one data point in each site at each time period. The red and blue lines connect the sites from the West Atlantic and the sites from the East Atlantic and MAR, respectively

during the YD decline of AMOC. Establishing the existence of such spatial differences needs to be constrained using additional cores with high sedimentation rates. Other strong evidence of YD weakening of AMOC have included $^{14}C$ records[3,41] (Fig. 3c), sortable silt grain size[6] (Fig. 3d) and geostrophic reconstruction of Gulf Stream strength[7]. Modelling studies have highlighted the role of AMOC reduction in causing widespread cooling of the northern hemisphere during the YD[45,48], such as that shown in the Greenland temperature record (Fig. 3f). The mechanism that could sustain a reduced AMOC during this stadial period is not well constrained. There is no evidence for meltwater discharge to the Bay of Biscay (Fig. 3a) suggesting limited Eurasian freshwater forcing. By contrast, there is some evidence of rerouting of Laurentide-sourced meltwater discharge to the Arctic Ocean at the onset of YD, but it was rather short-lived[49]. The Feni Drift IRD record indicates increased iceberg discharge from ~13 to 11 ka, but the magnitude and duration of these events were much smaller than those in HS1 (Fig. 3a). A potential explanation for these observations links to the stability of AMOC when northern ice sheets reached intermediate heights[5,32,42,43] such as those during the YD. Under an intermediate ice volume like YD, climate has been shown to be more variable[35] and AMOC has been found to be more sensitive to climate forcing[5], so freshwater forcing associated with the meltwater discharge and the melting icebergs described above might be sufficient to sustain the AMOC decline during this stadial period.

The compiled North Atlantic dataset shows a strong decrease in $^{231}Pa/^{230}Th$ towards present-day values during the early Holocene (Fig. 2). The inferred rebound of deep Atlantic circulation from the YD to early Holocene is associated with Greenland warming (Fig. 3f), and with the ventilation of deep-ocean carbon to the atmosphere as indicated by Atlantic $^{14}C$ observations[3] (Fig. 3c) and atmospheric $^{14}C$[50] and $CO_2$ records[51]

(Fig. 3e). Our $^{231}Pa/^{230}Th$ dataset suggests that deep-water transport in the Atlantic Ocean first reached its modern strength around 8 ka (Fig. 2), in agreement with water-mass proxy records[52,53], and consistent with the commencement of deep-water formation in the Labrador Sea[54] and the onset of stable present-day temperatures in the north[55].

An additional observation from the compiled dataset is an indication of contrasting behaviour between the East and West Atlantic over the last 25 kyr. The higher LGM $^{231}Pa/^{230}Th$ compared to late Holocene $^{231}Pa/^{230}Th$ observed in all western cores (Fig. 4) most likely indicates a weaker deep-water transport, consistent with sortable silt (10–63 μm) grain size data from the deep western boundary[56]. Supporting evidence for reduced rates of deep transport in the LGM Atlantic come from $^{14}C$ records[41], stable isotope modelling of oceanic vertical mixing[57], and geostrophic reconstruction of the Gulf Stream[7]. The picture emerging from the $^{231}Pa/^{230}Th$ observations at the Iberian margin (38° N, 3.14 km), off Mauritania (18° N, 3.09 km), and Sierra Leone Rise (7° N, 3.43 km) (Fig. 4) hints at a difference in deep-water transport at the mid-low latitudes of East Atlantic, with potential for stronger LGM transport in these deep eastern sites compared to the west. Brine rejection during ice formation in the Nordic seas has the potential to form deep waters that are not found at present[58], and this mechanism might provide a route for deep water overflows through the Iceland-Scotland Ridge into the eastern basin instead of via the shallower Denmark Strait into the western basin. This hypothesis is supported by a recent LGM ocean transport model, which simulates a deep-water layer (~2.5–3.5 km depth) which has a higher proportion of northern-sourced water in the eastern basin than the western basin[59]. Supporting evidence for a deep Nordic water mass in the North-East Atlantic comes from B/Ca and $\delta^{13}C$ ratios in the depth range of 2–3.5 km, between glacial northern-sourced

intermediate waters and AABW[60]. The $^{231}$Pa/$^{230}$Th observations at the eastern sites (3.1–4.3 km) potentially reflect the transition from the deep Nordic water to an increased influence of glacial AABW with depth (Fig. 4a). The persistently low $^{231}$Pa/$^{230}$Th (<0.058) observed in the east (Sierra Leone Rise, 7° N, 3.43 km) (Fig. 2c) supports the continual presence of deep-water transport exporting $^{231}$Pa from the low latitudes of the East Atlantic during the deglaciation, including HS1. It will be important to further examine the East-West difference suggested above, and proposed future work includes obtaining additional well-resolved $^{231}$Pa/$^{230}$Th time-series from the mid-altitude and low latitudes of the eastern basin.

In conclusion, we found that sedimentary $^{231}$Pa/$^{230}$Th time-series from the West and deep high-latitude North Atlantic provide compelling evidence of coherent changes in deep Atlantic circulation strength coincident in timing with millennial-scale climate changes during the deglacial period (Fig. 2). This finding is consistent with previous reconstructions of the strengths of Iceland-Scotland Overflow Water[6] (major component of NADW) and Florida Current[7] (part of the Gulf Stream: shallow northward flow with a component that compensates for Atlantic deep-water transport southward). The $^{14}$C ventilation records of deep and intermediate water masses in the North Atlantic[3,41] are also in agreement with our reconstructions. Together, these data support an overall reduced AMOC rate during HS1 and YD, with resumption during the BA, as previously proposed[2].

The remarkable consistency of the combined proxy records of ocean circulation rate has allowed us to provide a more in-depth analysis of the LGM-deglacial phasing of ocean circulation, ice sheet and climate changes. In particular, we have identified two phases of AMOC reduction during HS1 (Fig. 3, Supplementary Figs. 6, 7). The sequence of events proposed here is one that supports the mechanisms whereby an early AMOC decline (~19–16.5 ka) was initiated and sustained by freshwater forcing from the Eurasian ice sheets[37,38] and icebergs. Subsurface ocean warming in the Labrador Sea[27] caused by this AMOC slowdown triggered episodes of iceberg rafting from the Laurentide ice sheet from ~16.5 to 15 ka, leading to the second phase of maximum AMOC reduction (Fig. 3). This scenario explains the late arrival of the main ice-rafting event from the Laurentide ice sheet (~16.5–15 ka), characterised as the classic (sensu stricto) Heinrich Event 1[29,30].

The dataset presented in this study also hints at a different circulation history between the east and west basins of North Atlantic, with potential for stronger LGM deep-water transport in the east. This interpretation is in accord with previous recon-structions suggesting a weaker deep western boundary current[56] and the active southward transport of a Nordic deep-water mass (2–3.5 km) in the eastern basin[60] during the LGM. This East-West contrast potentially persisted over the ensuing deglaciation at the low latitudes, providing some evidence that deep circulation in the Atlantic Ocean did not cease during the cold stadial, despite an overall weakened AMOC.

The compilation of sedimentary $^{231}$Pa/$^{230}$Th records presented in this study also provides a refined dataset for future ocean transport models[13,14] to further constrain the magnitudes and rates of AMOC changes during millennial-scale climate events. Together with transient modelling efforts, a detailed picture of AMOC forcings and feedbacks during abrupt climate changes is emerging, providing insights into future long-term changes in AMOC behaviour associated with global climate change.

## Methods

**The four new sedimentary $^{231}$Pa/$^{230}$Th records.** New sedimentary $^{231}$Pa/$^{230}$Th data were collected on four deep sediment cores from the tropical Atlantic. Three of these cores are located to the west of the MAR: JC094-GVY14 (15.464° N, 50.992° W, 2.714 km water depth), EW9209-1JPC (5.907° N, 44.195° W, 4.056 km), and EW9209-3JPC (5.313° N, 44.26° W, 3.288 km), while the fourth is located to the east

of the MAR: JC094-GVY01 (7.435° N, 21.7963° W, 3.426 km) (Supplementary Fig. 8). Core 1JPC and 3JPC were retrieved at Ceara Rise in 1992[61] and have been stored in the core repository of Woods Hole Oceanographic Institution (WHOI). GVY14 and GVY01 are gravity cores recently retrieved at Researchers Ridge and Sierra Leone Rise, respectively, during the JC094 cruise which took place in October–November of 2013[62]. The $^{231}$Pa/$^{230}$Th analysis of these four cores provides new information on deep circulation (2.5–4 km) at the low latitudes of the Atlantic and enables a com-parison of lateral transport at ~3 km depth in the east and west basins.

The two new cores from Researchers Ridge and Sierra Leone Rise GVY14 and GVY01 were put in stratigraphic context using planktonic δ$^{18}$O data (Supplementary Fig. 9), and their age models for the last 25 kyr were developed using eleven and six $^{14}$C dates, respectively, (Supplementary Fig. 10, Supplementary Data 1). The existing age models for the Ceara Rise cores 1JPC and 3JPC[63,64] were also improved for this study, with five and nine new $^{14}$C dates, respectively (Supplementary Fig. 10, Supplementary Data 1).

Core 1JPC has HS1 $^{231}$Pa/$^{230}$Th measurements which are consistent with the picture painted by other western cores, but there is some chronological uncertainty over the HS1 interval (Supplementary Fig. 10), and therefore those data (Supplementary Fig. 12) are not included in the main results. Core GVY14 has low sedimentation rate (<1 cm ka$^{-1}$) over 20–15 ka (Supplementary Fig. 11) and thus lacks sufficient data resolution to document the HS1 shift in $^{231}$Pa/$^{230}$Th.

**Sediment chronology.** Errors associated with sediment chronology are critical when comparing the new and previously published $^{231}$Pa/$^{230}$Th time-series (Fig. 2, Supplementary Fig. 2). Therefore, we established the sediment chronologies of these cores using a consistent approach to provide the best possible age comparisons. All $^{14}$C ages were calibrated or re-calibrated against the atmospheric $^{14}$C curve after applying surface reservoir correction using OxCal version 4.2[65] and MarineCal13 calibration curve[50] to calculate the calendar ages of the samples. For the published records, we employed surface reservoir ages recommended by the original authors for the respective core sites. For the new cores, we used a constant surface reservoir age of 400 years (modern day global mean). We did not make changes to the non-$^{14}$C chronological tie-points (δ$^{18}$O, IRD count and correlation of sediment com-position) employed in published age models (Supplementary Table 1).

Age models for the sediment cores were developed or re-developed by interpolating ages between both the $^{14}$C and non-$^{14}$C chronological tie-points. The resulting age models are dependent on the methods employed for age interpolation[66]. Here, we briefly compared two such methods: OxCal Poisson deposition method which assumes random sedimentation rate with time defined by Poisson distribution[67], and a method assuming linear interpolation between the chronological tie-points[66]. The two methods yield some differences in the age models that do not appear to be significant for our study (Supplementary Fig. 13). The age models displayed in the main figures are those derived with OxCal Poisson deposition method.

Surface reservoir age variability is a significant source of uncertainty in $^{14}$C-based age models for core sites affected by sea ice melt during the stadials (HS1 and YD) and the LGM[68]. This surface reservoir uncertainty needs to be taken into account when considering the phasing of AMOC, meltwater discharge and iceberg calving during the last glacial termination (Fig. 3). Among the six $^{231}$Pa/$^{230}$Th time-series (Fig. 3b) used to interpret the phasing of AMOC and ice system changes, the age model for the Rockall basin core (50° N, 4.28 km) was not developed with $^{14}$C-derived chronological tie-points[16]. The other five $^{231}$Pa/$^{230}$Th records with $^{14}$C-derived chronological tie-points are located at lower latitudes (<35° N/S) (Fig. 3b), with mean surface reservoir ages of 400 ± 400 years over the last 25 kyr[69]. Sediment records of fluvial discharge and IRD discussed in this study are located at higher latitudes (>45° N) (Fig. 3a), and substantial increases in surface reservoir ages during HS1 and YD (>1000 years) have been found at these locations[69]. Here, we test the significance of reservoir age uncertainty by computing several sets of sediment core age models using maximum and minimum surface reservoir values expected for high latitudes (>45° N) and low latitudes (<35° N/S) over the last 25 kyr[69]. In the different scenarios, we found shifts in the absolute timing of AMOC, meltwater discharge and IRD events (Supplementary Fig. 14), but these shifts do not affect our main interpretations. In particular, the beginning of AMOC decline at early deglacial still coincides with intensified Eurasian meltwater discharge, and Laurentide-sourced IRD is only evident later at the peak of AMOC reduction (Supplementary Fig. 14). The age models displayed in the main figures are those developed using mean surface reservoir values derived for high latitudes (>45° N) and low latitudes (<35° N/S) over the last 25 kyr[69].

**Analytical techniques.** Sedimentary $^{231}$Pa/$^{230}$Th analysis was carried out by measuring U, Th and Pa isotopes in the bulk sediment (Supplementary Data 2) using isotope dilution[70]. Some of the U-series measurements for the two Ceara Rise cores 1JPC and 3JPC were carried out at the Woods Hole Oceanographic Institution (WHOI) on a Thermo-Finnigan Element 2 single collector, inductively coupled plasma-mass spectrometer (ICP-MS)[2], and at Lamont-Doherty Earth Observatory (LDEO) using a Thermo Scientific Element XR, single collector, inductively coupled plasma-mass spectrometer (ICP-MS)[2]; and the remainder of the measurements were made using a new protocol set up at the Bristol Isotope Group lab of Uni-versity of Bristol (UoB) (Supplementary Fig. 12). The new UoB protocol had been demonstrated to yield U-series isotope measurements of good precision and

accuracy for sedimentary $^{231}$Pa/$^{230}$Th analysis by producing 10 replicate measurements (average $^{231}$Pa: 1.24 ± 0.04 dpm g$^{-1}$, 230Th: 5.48 ± 0.09 dpm g$^{-1}$) of a homogenised Southern Ocean siliceous ooze which agree with the measurements made in GEOTRACES Th and Pa intercalibration study (reference value $^{231}$Pa: 1.25 dpm g$^{-1}$, $^{230}$Th: 5.53 dpm g$^{-1}$)[71]. The UoB procedures for sample preparation consists of $^{236}$U, $^{229}$Th and $^{233}$Pa spike addition, sediment digestion, co-precipitation of actinide elements with Fe hydroxide and chemical separation of the U, Th and Pa via ion-exchange chromatography. Sample analyses at UoB were carried out using a Thermo-Finnigan Neptune, multicollector, ICP-MS. The methods employed followed those in Burke and Robinson (2012)[72] for U and Auro et al. (2012)[73] for Th and Pa, with $^{229}$Th and $^{230}$Th measured alternately on the Secondary Electron Multiplier (SEM), and $^{231}$Pa and $^{233}$Pa measured on a multi-ion counter (MIC) array to avoid significant build-up of dark noise on the SEM. Sedimentary $^{231}$Pa$_{ex}$/$^{230}$Th$_{ex}$ was calculated by correcting the $^{231}$Pa and $^{230}$Th measurements for fractions that are supported by the decay of lithogenic and authigenic U by assuming a lithogenic $^{238}$U/$^{232}$Th activity ratio of 0.6. We found that the choice of lithogenic $^{238}$U/$^{232}$Th within the uncertainty of the derived Atlantic average value (0.6 ± 0.1, or more recently, 0.55 ± 0.16)[8,74] made no significant difference to the $^{231}$Pa/$^{230}$Th values for our four new cores (example in Supplementary Fig. 12). The $^{231}$Pa and $^{230}$Th measurements were also corrected for radioactive decay to the age of sediment deposition. The disequilibria that might result from alpha recoil[74] was not explicitly taken into account in the $^{231}$Pa/$^{230}$Th calculations, as we did not have an independent measure of this effect. Core 3JPC had four $^{231}$Pa/$^{230}$Th measurements published in a previous study[12], and the values were re-calculated (Supplementary Fig. 12) to correct for radioactive decay using the revised sediment ages. Uncertainty in $^{231}$Pa/$^{230}$Th measurements was propagated from internal errors associated with weighing, spike calibration and ICP-MS measurements using a Monte-Carlo method.

Sediment biogenic opal content was analysed using the alkaline extraction method and molybdate-blue spectrophotometry following the procedures in Mortlock and Froelich[75]. The vertical flux of bulk sediment was determined using the $^{230}$Th-normalisation method, which assumes that the rate of removal of $^{230}$Th onto particle surfaces is equal to the production rate of $^{230}$Th in the water column[76]. The vertical opal flux was calculated by multiplying $^{230}$Th-normalised bulk sediment flux by the fraction of opal in the sediment.

**Data availability**. The data reported in this paper are listed in the Supplementary Information and archived in Pangaea database (https://doi.org/10.1594/PANGAEA.890942).

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

## Acknowledgements

We would like to express our gratitude to the European Research Council and NERC Radiocarbon Facility for providing support to this study. The contribution of Jerry McManus was supported in part by the US-National Science Foundation and the Comer Science and Education Foundation. Ruza Ivanovic was supported by NERC grant NE/K008536/1. We want to thank Susan Brown-Leger and Marti Jeglinski for helping with $^{231}$Pa/$^{230}$Th sample preparation and analysis at WHOI, and Marty Fleisher and L. Gene Henry for assistance at LDEO. We appreciate the help and advice from Tim Elliot, Chris Coath, Katherine Adena and Carolyn Taylor from the Bristol Isotope Group in setting up the sediment $^{231}$Pa/$^{230}$Th analytical protocol at the UoB. We thank Ellen Roosen for help with sampling of sediment cores 3JPC and 1JPC at the WHOI core repository. Great thanks to the scientists and crew of JC094 for their assistance in collection of GVY01 and GVY14 sediment core from the tropical Atlantic Ocean. We also thank the undergraduate volunteer students/alumni from the UoB (Sam Lucas, Bryony Essex, Kasia Clarke) for their assistance in sample preparation.

## Author contributions

H.C.N., L.F.R., K.J.M. and A.W.J retrieved GVY01 and GVY14 sediment cores during the JC094 cruise and developed the age models for the two new cores. J.F.M provided some of the U-series measurements for the EW9209-1JPC and EW9209-3JPC sediment cores at the LDEO, and the remaining measurements were carried out at the UoB by H.C.N. All authors (H.C.N., L.F.R., J.F.M., K.J.M., A.W.J., R.F.I., L.J.G. and T.C.) contributed to the data interpretation and participated in the manuscript preparation.

## Additional information

**Competing interests:** The authors declare no competing interests.

