## [Peer Review File · Nature Communications]

Reviewers' comments:

Reviewer #1 (Remarks to the Author):

Review:

Despite the enormous effort needed for obtaining $^{231}\text{Pa}/^{230}\text{Th}$ I consider this proxy as one of the most valuable in paleoceanography, because it can provide unique information not available from other proxies, when measured correctly and interpreted carefully. Here Ng et al. present a new data set of four $^{231}\text{Pa}/^{230}\text{Th}$ down-core profiles. As mentioned before, measuring $^{231}\text{Pa}/^{230}\text{Th}$ is a laborious task and not easy at all. Therefore I very much welcome the presentation of ~ 70 new $^{231}\text{Pa}/^{230}\text{Th}$ data points. This is hard won data and the authors, in particular the lead author, deserve respect for the hard lab work.

The new data sets are compared with existing down-core profiles, however, not with all of which are available from the literature (this will be pointed out later, #3).

It is highly appreciated that the raw data of both the $^{231}\text{Pa}/^{230}\text{Th}$ and the ^{14}C is provided in the supplement. In general the manuscript is nicely written.

Here the authors claim to provide new insights into the timing and spatial distribution of AMOC changes "[...] by assessing the details of why, how, when, and where the $^{231}\text{Pa}/^{230}\text{Th}$ changes occurred." I think here they are little bit too optimistic and I don't believe they comprehensively can answer the questions for "why, how, when, and where" from the new data and their approach. I also do not think that the specific conclusions of this paper revolutionize our view on past ocean circulation, in particular on the deglacial Atlantic. The approach of compiling $^{231}\text{Pa}/^{230}\text{Th}$ data for the deglacial Atlantic is more or less a sophisticated remake of the approaches already published by (Bradtmiller et al., 2014; Gherardi et al., 2009; Lippold et al., 2016) (not mentioning the legion of other papers dealing with the deglacial AMOC based on other proxies). However, they also provide some new ideas (at least new to me) which have not been shown before by means of $^{231}\text{Pa}/^{230}\text{Th}$: (1) the different behaviour of the circulation in the East and the West Atlantic and (2) the inner-HS1 variations.

Specific comments:

1. Continuing with the abovementioned novel main messages of the paper I wonder if it might be more reasonable including the east-west gradient and/or the inner-HS1 structure into the title rather than "coherent deglacial changes", which are neither novel nor that clear in my opinion.
2. The quality of $^{231}\text{Pa}/^{230}\text{Th}$ is very good. Thanks to the provision of the raw data I was able to recalculate the values and I came to the same result. Only the choice of the lithogenic contributions to excess ^{231}Pa and ^{230}Th should be better justified. 0.6 seems to me unfounded, because bulk $^{238}\text{U}/^{232}\text{Th}$ is sometimes lower. So 0.5 would be more appropriate (see approach by (Bourne et al., 2012)). However, I realized that values different from 0.6 do not change the $^{231}\text{Pa}/^{230}\text{Th}$ values. So this is not a crucial point. Was the recoil effect on ^{230}Th considered?
3. One point I clearly do not agree, and which requires correction, is the choice of down-core profiles creating the "composite record" for two reasons:
 - a) the "composite record" is composed from locations of different latitudes and different water depths. But merging absolute $^{231}\text{Pa}/^{230}\text{Th}$ values from different water depths within the circulation cell of a distinct water mass is an invalid approach. $^{231}\text{Pa}/^{230}\text{Th}$ tends to decrease with increasing water depth within a circulation cell (e.g. (Burckel et al., 2016)). Therefore, averaging absolute values from different water depths is unwise. The same problem appears when merging different latitudes, even though the water depth effect is more important (Marchal et al., 2000). Thus, combining $^{231}\text{Pa}/^{230}\text{Th}$ values from 2.71 km with 4.06 km does not make sense. I think this can be seen very nicely from Fig. 16 in the paper by (Luo et al., 2010): A single absolute $^{231}\text{Pa}/^{230}\text{Th}$ value for one location is not representative for the AMOC state of a distinct time period, but can be interpreted in the context of other locations only.
 - b) Although I appreciate the nice overview of cores, given in Supplementary table 1, I cannot follow the selection procedure for choosing the $^{231}\text{Pa}/^{230}\text{Th}$ down-core profiles from the literature to be considered here. The authors already ignore a number of cores due to co-variations with opal, which is a questionable approach (will be pointed out later), but they miss also a large number of published profiles, which should not be neglected. E.g.: MD95-2037 (Gherardi et al.,

2009), MD03-2705 (Meckler et al., 2013), MD02-2594 and MD02-2588 (Negre et al., 2010), IODP1313 (Lippold et al., 2016), GeoB3808-6 (Jonkers et al., 2015), GeoB16202-2 (Mulitza et al., 2017), DACP2 (Hall et al., 2006), BOFS 17K (Roberts et al., 2014), GeoB9508-1 (Lippold et al., 2012). I'm quite confused here, because I cannot identify the systematics behind the selection. Sometimes a core is considered while another from the same publication is not. This needs to be improved.

4. $^{231}\text{Pa}/^{230}\text{Th}$ -data from EW9209 3JPC has been already reported by (Bradt Miller et al., 2014). Why are the values from this paper not included here? Maybe I'm wrong, but I feel there is a lack of consistency. Or is this maybe just an issue of cutting decimal places? The (Bradt Miller et al., 2014) data would be:

mcd	ka	Pa/Th
0.06	1.59	0.047
0.76	17.10	0.075
0.96	19.33	0.071
1.06	21.13	0.060

But these values are not visible or slightly different in this manuscript.

5. The authors did not consider $^{231}\text{Pa}/^{230}\text{Th}$ profiles when these show correlations to opal. However, the here applied criteria seem arbitrary to me. I would not disregard time series of $^{231}\text{Pa}/^{230}\text{Th}$ simply because there is a correlation coefficient higher than $r=0.6$ with opal. Why 0.6? And even if so, why should this be a problem? When looking at Supplementary Figure 1a I see a cloud of values clustered between 0 and 0.15 $\text{gcm}^{-2} \text{ k}^{-1}$ and from 0.02 to 0.13 in $^{231}\text{Pa}/^{230}\text{Th}$ with one exception (red crosses core 5). For me this indicates that there is no basin-wide correlation of $^{231}\text{Pa}/^{230}\text{Th}$ with opal as long as opal is not too high (e.g. $\text{gcm}^{-2} \text{ k}^{-1}$), as can be seen in the Southern Ocean. So, what could be a reason for correlations of opal with $^{231}\text{Pa}/^{230}\text{Th}$ for individual cores? I think it is obvious that $^{231}\text{Pa}/^{230}\text{Th}$ increases when a location is bathed by sluggish AABW instead of vigorous NADW. But AABW also hold higher concentrations of silicate and thus sustains higher levels of opal preservation. When neglecting all this locations important paleoceanographic insight will be lost. That's why I do not agree to the statement in line 42, disregarding valuable data sets for interpreting basin wide $^{231}\text{Pa}/^{230}\text{Th}$, which should be revised accordingly.

6. I would have found it more helpful if the core names would have been given in the figures instead of numbers and latitudes. I'm aware that there is a space issue, but reading would be much easier with core names indicated.

7. The main finding of the difference between east and west circulation during LGM should be elaborated better. There is little evidence from the east (especially when disregarding most of the available profiles there). What is the reason for the very different shape of the profiles from cores GUY01 and RC24-12 compared to SU81-18? And is this number of cores sufficient to allow deriving the east-west gradient in the LGM?

8. line 79: please insert "mainly" between "is" and "derived", because there are unavoidably contributions from above. These are just smaller, due to the increasing concentrations with water depth.

9. line 124: values "[...] range from 0.059 to 124 0.083 and are higher overall than the Holocene (<10 ka) values, which range from 0.041 to 0.065.". Given these wide ranges within one time period from different site, does it really make sense to create a "composite record"?

10. line 146: although I find the idea of an different AMOC-state at the West and East during the LGM appealing, I wonder if one core (SU81-18) from the North-Atlantic is sufficient to support this hypothesis.

11. line 160: I would rephrase this sentence, because this has been proposed already 13 years before (McManus et al., 2004).

12. line 207: I'm not sure what Fig. 4 can tell us here, with most of existing data sets missing. A reduced deep-water transport during the LGM is not new. But how can this be driven by the Labrador Sea Water, which is rather found in the upper NADW?

13. line 240: it would be interesting to calculate how long the cessation of strong NADW formation must be, that it can be recorded by $^{231}\text{Pa}/^{230}\text{Th}$.

14. line 259: "Gulf Stream" is an incorrect oversimplification.

15. line 327: Although the authors are well aware of the problems regarding reservoir ages during HS1, they still stick with 400 years. However, the here intrinsic implemented uncertainty endangers any conclusion on trigger and consequence of iceberg (meltwater) purges. The here suggested sequence of events (Eurasian Ice sheets→AMOC reduction→subsurface warming→Laurentide ice sheets→second phase of AMOC reduction), which is contradictive to the scenario proposed by (Barker et al., 2015) is highly dependent on the timing. I wonder if the sequence of events can be maintained with reservoir ages >400 years. Maybe the authors can go through the scenario of higher reservoir ages and test if this conclusion would be still valid.

16. line 354: the certified and measured references values should be given and would be worth a diagram in the supplement.

References

- Barker, S., Chen, J., Gong, X., Jonkers, L., Knorr, G., Thornalley, D., 2015. Icebergs not the trigger for North Atlantic cold events. *Nature* 520, 333-336.
- Bourne, M., Thomas, A., Niocaill, C., Henderson, G., 2012. Improved determination of marine sedimentation rates using ^{230}Th s. *Geochemistry Geophysics Geosystems* 13, Q09017.
- Bradt Miller, L., McManus, J.F., Robinson, L.F., 2014. $^{231}\text{Pa}/^{230}\text{Th}$ evidence for a weakened but persistent Atlantic meridional overturning circulation during Heinrich Stadial 1. *Nature Communications* 5, 5817.
- Burckel, P., Waelbroeck, C., Luo, Y., Roche, D., Pichat, S., Jaccard, S.L., Gherardi, J., Govin, A., Lippold, J., Thil, F., 2016. Changes in the geometry and strength of the Atlantic Meridional Overturning Circulation during the last glacial (20-50 ka). *Climate of the Past* 12, 2061–2075.
- Gherardi, J., Labeyrie, L., Nave, S., Francois, R., McManus, J., Cortijo, E., 2009. Glacial-interglacial circulation changes inferred from $^{231}\text{Pa}/^{230}\text{Th}$ sedimentary record in the North Atlantic region. *Paleoceanography* 24, PA2204.
- Hall, I., Moran, S., Zahn, R., Knutz, P., Shen, C., Edwards, R., 2006. Accelerated drawdown of meridional overturning in the late-glacial Atlantic triggered by transient pre-H event freshwater perturbation. *Geophysical Research Letters* 33, L16616.
- Jonkers, L., Zahn, R., Thomas, A., Henderson, G., Abouchami, W., François, R., Masque, P., Hall, I.R., Bickert, T., 2015. Deep circulation changes in the central South Atlantic during the past 145 kyrs reflected in a combined $^{231}\text{Pa}/^{230}\text{Th}$, Neodymium isotope and benthic record. *Earth and Planetary Science Letters* 419, 14-21.
- Lippold, J., Gutjahr, M., Blaser, P., Christner, E., Ferreira, M.-L.C., Mulitza, S., Christl, M., Wombacher, F., Böhm, E., Antz, B., Cartapanis, O., Vogel, H., Jaccard, S., 2016. Deep water provenance and dynamics of the (de)glacial Atlantic meridional overturning circulation. *Earth and Planetary Science Letters* 445, 68-78.
- Lippold, J., Mulitza, S., Mollenhauer, G., Weyer, S., Christl, M., 2012. Boundary scavenging at the east Atlantic margin does not negate use of Pa/Th to trace Atlantic overturning. *Earth and Planetary Science Letters* 333–334, 317-331.
- Luo, Y., Francois, R., Allen, S., 2010. Sediment $^{231}\text{Pa}/^{230}\text{Th}$ as a recorder of the rate of the Atlantic meridional overturning circulation: insights from a 2-D model. *Ocean Science* 6, 381-400.
- Marchal, O., Francois, R., Stocker, T., Joos, F., 2000. Ocean thermohaline circulation and sedimentary $^{231}\text{Pa}/^{230}\text{Th}$ ratio. *Paleoceanography* 15, 6.
- McManus, J., Francois, R., Gherardi, J., Keigwin, L., Brown-Leger, S., 2004. Collapse and rapid resumption of Atlantic meridional circulation linked to deglacial climate change. *Nature* 428, 834-837.
- Meckler, A.N., Sigman, D.M., Gibson, K.A., Francois, R., Martinez-Garcia, A., Jaccard, S.L., Rohl, U., Peterson, L.C., Tiedemann, R., Haug, G.H., 2013. Deglacial pulses of deep-ocean silicate into the subtropical North Atlantic Ocean. *Nature* 495, 495-498.
- Mulitza, S., Chiessi, C.M., Schefuß, E., Lippold, J., Wichmann, D., Antz, B., Mackensen, A., Paul, A., Prange, M., Rehfeld, K., Werner, M., Bickert, T., Frank, N., Kuhnert, H., Lynch-Stieglitz, J., Portilho-Ramos, R.C., Sawakuchi, A.O., Schulz, M., Schwenk, T., Tiedemann, R., Vahlenkamp, M., Zhang, Y., 2017. Synchronous and proportional deglacial changes in Atlantic meridional overturning and northeast Brazilian precipitation. *Paleoceanography* 32, 622-633.

Negre, C., Zahn, R., Thomas, A., Masque, P., Henderson, G., Martinez-Mendez, G., Hall, I., Mas, J., 2010. Reversed flow of Atlantic deepwater during the Last Glacial Maximum. *Nature* 468, 84 - 89.

Roberts, N., McManus, J., Piotrowski, A., McCave, N., 2014. Advection and scavenging controls of Pa/Th in the northern NE Atlantic. *Paleoceanography* 29, 668–679.

Reviewer #2 (Remarks to the Author):

In the manuscript "Coherent deglacial changes in deep Atlantic Ocean circulation", Ng et al reconstruct AMOC variation in the last deglaciation by sorting 10 out of 23 Pa/Th records and try to provide a basin-wide constraint on the timing and magnitude of the deglacial circulation changes after adjusting chronology. I'm not expert on the proxy, but no surprise to me, even based on a composite dataset the outcome generally falls in line with previously published reconstructions and it seems that the new Pa/Th dataset cannot well capture the abruptness of AMOC recovery during abrupt BA warming. Is this due to the smoothing choice or the composite approach itself? In addition, the mechanisms accounting for the AMOC slow-down during the early stage of HS1 and YD are not novel and bit unconvincing. At least, a comprehensive discussion about potential mechanisms, perhaps supported by modeling, is important and necessary for publication in *Nature Communications*. Based on its current quality, I would suggest at least a major revision before acceptance. Comments below may help improve it.

1. Regarding triggering mechanisms of AMOC slow-down during deglaciation,

a. the authors try to use a single IDR record from Feni Drift to argue that the early deglaciation of FIS could be the trigger of the first phase of HS1 (~18.5-17ka). But why there is no AMOC slow down during ~21ka BP when there was also a robust IRD release events? See also some similar paper like Peck 2006, EPSL, Toucanne et al 2015 QSR. They also show pre-deglaciation IRD/drainage events, but AMOC is still active at that moment according to Pa/Th. Does the sensitivity of the ocean circulation depend on background climate? This needs to be well discussed.

b. That the authors repeat the ice-sheet melting theory to interpret the YD based on a single IRD record, seems to me that they did not think about this issue seriously. From my point of view, freshwater forcing can be a potential trigger for YD, although the triggering mechanism remains elusive (e.g. Carlson and Clark 2012 *Rev Geo*; Condron and Winsor 2012 *PNAS*). One fundamental feature is that the sea level rise is just ~ 5m during the YD interval. Given the available evidence from reconstructions and climate modeling, it seems to me that the discrepancy among different hypothesis could be potentially reconciled if the climate system is in a window of AMOC bistability during the YD (e.g. Zhang et al 2014, 2017 *NGS*). In such a window, a short-lived freshwater perturbation can cause an AMOC slow-down for a long time that can be longer than the duration of the perturbation itself.

2. Regarding the two circulation regimes between high latitudes-western Atlantic and eastern Atlantic, the authors employed the GIS ridge overflow to interpret the low Pa/Th value (strong circulation) in the core (34N, 3.14km). If so, why the high-latitude eastern core (50N, 4.28km) that is much closer to the GIS ridge would characterize a high value (weaker circulation)? Interestingly, the cores in the eastern side indicate that the deeper the records are the weaker circulation will be. Thus, it is not proper to conclude "a fundamental deglacial reorganization of deep water formation" based on the shallowest record since the rest eastern records indicate a weak circulation during the LGM in comparison to Holocene.

In general, the core (34N, 3.14km) shows an abnormal feature than the rest, potentially indicating other processes involved, for instance, the NA subtropical circulation. In the North Atlantic, the deep ocean currents are strong in the western (i.e. deep western boundary current) and weak in the eastern. Given the barotropic feature of the subtropical gyre, it is plausible that the record includes a signal from subtropical circulation. According to several modeling studies (e.g. Gong et

al 2015 Clim Dyn), it appears that higher northern hemisphere ice sheet can lead to a stronger subtropical circulation. In addition, the Pa/Th value does indicate a weaker circulation during the Holocene than the LGM. Thus, it is plausible that the core (34N, 3.14km) mainly records the strength of the subtropical circulation, rather the deep ocean current. Since several modelers are co-authored in this study, it should be easy to test it by new/published modeling results.

Reviewer #3 (Remarks to the Author):

Review of Ng et al.,

Ng and co-authors present 4 new Pa/Th reconstructions from the Atlantic Ocean, which cover the last deglaciation. These new cores are added to previously published cores to provide a composite of Pa/Th variations across the deglaciation.

As Pa/Th is a proxy for deep water transport, AMOC changes from the LGM to the Holocene are discussed.

The Pa/Th composite is a welcome addition and nicely confirms previous assumptions about AMOC changes across the deglaciation.

Records are also analyzed with respect to their zonal position in the Atlantic. It is then suggested that deep water transport was a bit stronger in the Eastern part of the Atlantic than in the Western part of the Atlantic at the LGM and until the Holocene.

While I think that the Pa/Th composite is very nice and needed addition, the manuscript could be improved.

Please find below some specific comments on the manuscript.

1) The introduction is focused on Pa/Th and challenges associated with it being a proxy for deep water transport. Little to no scientific rationale in generating a Pa/Th composite across the deglaciation is presented. While this would be fine for a more focused journal, I feel that for a journal such as Nature Communication, the introduction should focus on the scientific question to be answered.

2) The "Results" section is extremely short and in fact just gives a very broad overview of Pa/Th changes across the deglaciation: i) overview of Pa/Th values at different times of the deglaciation without any specific context, ii) two phases changes during HS1, iii) East-West contrast. Please note that p7, L. 145-146 it is noted that the Rockall plateau Pa/Th values are similar at the LGM and Holocene, which does not seem to be the case in figs 2 and 4...

3) The discussion starts with HS1, which is very interesting and most likely the most interesting part of the ms as also highlighted in the abstract. But then it goes onto discussing the East-West gradient across the last 25 kyrs. This different behavior zonally across the Atlantic is not as robust as the rest and, if anything, is mostly seen at the LGM. As such most of the discussion focuses on the LGM. This gives rise to 2 issues:

i) The discussion starts with HS1, moves to the LGM and then fairly quickly across the deglaciation with most events mentioned briefly in one paragraph. This break in the chronology is a bit distracting and all the succession of BA, YD, Holocene reads like a list with approximation.

ii) The zonal difference relies mostly on one previously published core: SU81-18 from Gherardi et al., 2005.

The data from the Iberian margin and the Sierra Leone Rise lend support to the zonal difference at the LGM. However, I'm not sure the evidence for zonal differences during the deglaciation is robust. The Iberian margin core nicely displays H1, BA, YD variations, which seem to follow the composite pa/Th very closely: i.e. I'm not sure the East-West difference is statically significant outside of the LGM.

Cores 7 and 10 are somewhat lower resolution, their variations in general more muted. These cores are "quite far" from the NADW path, which flows as a western boundary current in the deep

Atlantic. These cores could be influenced by changes in the upwelling system instead.

Figures: The figures are very nice and informative.

Figure 1:

An useful system of symbols and numbers is set up to identify cores. This is very nice, but core names and references should at least be added here somewhere either on the figure or in the legend. Else the supplementary table should be moved to the main text.

Minor points:

-p2, L. 38: "relies"

-p7, L. 147: no coordinates are given for the Sierra Leone Rise core

-p9, L. 195: maybe "mid HS1" might be more appropriate than "late HS1".

-p10, L. 230; "These rate proxy records..." needs rephrasing

p12, L. 262: "This data supports"

- P12, L.266-267: I am not sure "an in-depth analysis of the LGM-deglacial phasing of ocean circulation, ice-sheet and climate changes" was given in the ms.

Response to reviewers

Key:

- Reviewers' comments
- Author's response

Reviewer #1 (Remarks to the Author):

Despite the enormous effort needed for obtaining $^{231}\text{Pa}/^{230}\text{Th}$ I consider this proxy as one of the most valuable in paleoceanography, because it can provide unique information not available from other proxies, when measured correctly and interpreted carefully. Here Ng *et al.* present a new data set of four $^{231}\text{Pa}/^{230}\text{Th}$ down-core profiles. As mentioned before, measuring $^{231}\text{Pa}/^{230}\text{Th}$ is a laborious task and not easy at all. Therefore I very much welcome the presentation of ~70 new $^{231}\text{Pa}/^{230}\text{Th}$ data points. This is hard won data and the authors, in particular the lead author, deserve respect for the hard lab work.

Reply #1: We thank the reviewer for their recognition of both the extensive effort and value represented by our new datasets. In the following we have tried to address each specific point raised.

The new data sets are compared with existing down-core profiles, however, not with all of which are available from the literature (this will be pointed out later, #3).

Reply #2: Please see Reply #8.

It is highly appreciated that the raw data of both the $^{231}\text{Pa}/^{230}\text{Th}$ and the ^{14}C is provided in the supplement. In general the manuscript is nicely written.

Here the authors claim to provide new insights into the timing and spatial distribution of AMOC changes “[...] by assessing the details of why, how, when, and where the $^{231}\text{Pa}/^{230}\text{Th}$ changes occurred.” I think here they are little bit too optimistic and I don't believe they comprehensively can answer the questions for “why, how, when, and where” from the new data and their approach.

Reply #3: We have removed the line in the revised manuscript.

I also do not think that the specific conclusions of this paper revolutionize our view on past ocean circulation, in particular on the deglacial Atlantic. The approach of compiling $^{231}\text{Pa}/^{230}\text{Th}$ data for the deglacial Atlantic is more or less a sophisticated remake of the approaches already published by (Bradt Miller *et al.*, 2014; Gherardi *et al.*, 2009; Lippold *et al.*, 2016) (not mentioning the legion of other papers dealing with the deglacial AMOC based on other proxies).

Reply #4: Even with the large efforts to understand deglacial circulation, there remain gaps. Our latest \$^{231}\text{Pa}/^{230}\text{Th}\$ time-series compilation is a significant advance from previous studies, because by combining all the available well-resolved records on a uniform age scale, we have been able to demonstrate that there are robust millennial-scale trends of deglacial AMOC changes that are not

dependent on single core location, especially those in the western basin. This alleviates the widespread concern of over-interpretation of unique sites. Of course, through this effort we have also highlighted potential new features of the deglacial circulation, as the reviewer acknowledges below.

However, they also provide some new ideas (at least new to me) which have not been shown before by means of $^{231}\text{Pa}/^{230}\text{Th}$: (1) the different behaviour of the circulation in the East and the West Atlantic and (2) the inner-HS1 variations.

Specific comments:

1. Continuing with the abovementioned novel main messages of the paper I wonder if it might be more reasonable including the east-west gradient and/or the inner-HS1 structure into the title rather than “coherent deglacial changes”, which are neither novel nor that clear in my opinion.

Reply #5: The inner-HS1 structure and East-West contrast are indeed new insights that have arisen from the study. However, we think that the remarkable coherence of the western $^{231}\text{Pa}/^{230}\text{Th}$ records will be the result of widest interest to the scientific community, including modellers and paleoceanographers. As such we suggest a minor change to the title of the paper – Coherent deglacial changes in western Atlantic Ocean circulation.

2. The quality of $^{231}\text{Pa}/^{230}\text{Th}$ is very good. Thanks to the provision of the raw data I was able to recalculate the values and I came to the same result. Only the choice of the lithogenic contributions to excess ^{231}Pa and ^{230}Th should be better justified. 0.6 seems to me unfounded, because bulk $^{238}\text{U}/^{232}\text{Th}$ is sometimes lower. So 0.5 would be more appropriate (see approach by (Bourne et al., 2012)). However, I realized that values different from 0.6 do not change the $^{231}\text{Pa}/^{230}\text{Th}$ values. So this is not a crucial point.

Reply #6: As the reviewer points out, the small difference that the choice of lithogenic $^{238}\text{U}/^{232}\text{Th}$ has on the $^{231}\text{Pa}/^{230}\text{Th}$ values means that this is not a crucial factor. We have continued to use 0.6 – which is the mean Atlantic value by Henderson and Anderson (2003) in the revised manuscript. We have added an example of changing the ratio of lithogenic $^{238}\text{U}/^{232}\text{Th}$ as a sensitivity test in Supplementary Fig. 10.

Was the recoil effect on ^{230}Th considered?

Reply #7: The disequilibria that may result from alpha recoil is not explicitly taken into account in our calculations, as we do not have an independent measure of this effect, but there is negligible difference in correcting the $^{231}\text{Pa}/^{230}\text{Th}$ data for alpha recoil by assuming lithogenic $^{234}\text{U}/^{238}\text{U}$, $^{230}\text{Th}/^{234}\text{U}$, $^{231}\text{Pa}/^{235}\text{U}$ of 0.96 (Bourne et al., 2012) instead of 1 (secular equilibrium).

3. One point I clearly do not agree, and which requires correction, is the choice of down-core profiles creating the “composite record” for two reasons:

a) the “composite record” is composed from locations of different latitudes and different water depths. But merging absolute $^{231}\text{Pa}/^{230}\text{Th}$ values from different water depths within the circulation cell of a distinct water mass is an invalid approach. $^{231}\text{Pa}/^{230}\text{Th}$ tends to decrease with increasing water depth within a circulation cell (e.g.(Burckel et al., 2016)). Therefore,

averaging absolute values from different water depths is unwise. The same problem appears when merging different latitudes, even though the water depth effect is more important (Marchal et al., 2000). Thus, combining $^{231}\text{Pa}/^{230}\text{Th}$ values from 2.71 km with 4.06 km does not make sense. I think this can be seen very nicely from Fig. 16 in the paper by (Luo et al., 2010): A single absolute $^{231}\text{Pa}/^{230}\text{Th}$ value for one location is not representative for the AMOC state of a distinct time period, but can be interpreted in the context of other locations only.

Reply #8: We agree that $^{231}\text{Pa}/^{230}\text{Th}$ changes with latitude and water depth. However, the purpose of our composite record is not to compute a single absolute $^{231}\text{Pa}/^{230}\text{Th}$ value for the whole (deep North) Atlantic at any single point of time (which would be an impossible task), but to show the average trends and amplitudes. Despite the depth and latitudinal variations, all records from the west and deep high latitude North Atlantic still display coherent $^{231}\text{Pa}/^{230}\text{Th}$ trends over the last 25 kyr (Fig. 2a), which suggests that our approach of constructing a composite record from these cores to reflect overall large-scale changes in AMOC strength is appropriate. Converting this composite record to an overall 'rate' or water mass transport would require more detailed modelling. For the above reasons, we decide to keep the composite record in the revised manuscript, as well as showing the $^{231}\text{Pa}/^{230}\text{Th}$ data from individual records that were used to construct the composite (Fig. 3b).

b) Although I appreciate the nice overview of cores, given in Supplementary table 1, I cannot follow the selection procedure for choosing the $^{231}\text{Pa}/^{230}\text{Th}$ down-core profiles from the literature to be considered here. The authors already ignore a number of cores due to co-variations with opal, which is a questionable approach (will be pointed out later), but they miss also a large number of published profiles, which should not be neglected. E.g.: MD95-2037 (Gherardi et al., 2009), MD03-2705 (Meckler et al., 2013), MD02-2594 and MD02-2588 (Negre et al., 2010), IODP1313 (Lippold et al., 2016), GeoB3808-6 (Jonkers et al., 2015), GeoB16202-2 (Mullitza et al., 2017), DAPC2 (Hall et al., 2006), BOFS 17K (Roberts et al., 2014), GeoB9508-1 (Lippold et al., 2012). I'm quite confused here, because I cannot identify the systematics behind the selection. Sometimes a core is considered while another from the same publication is not. This needs to be improved.

Reply #9: We made every effort to include all available cores – and some were initially excluded in the original manuscript as they were at shallower intermediate depths. However, in response to the reviewer's comment we have decided to include all intermediate-depth cores as well – as can be seen in the supplements. In the revised manuscript, we have added MD95-2037, MD02-2594, IODP1313, GeoB16202-2, DAPC2, BOFS 17K, GeoB9508-1, and the more recent GeoB16206-1 (Voigt et al., 2017) $^{231}\text{Pa}/^{230}\text{Th}$ profiles. The following cores have not been included for the following reasons:

a) MD03-2705: there is ^{230}Th data but not ^{231}Pa data.

b) MD02-2588: it is in the Indian Ocean.

c) GeoB3808-6: the data resolution is too low to be used in this study – there are only five data points over 25–0 ka.

4. $^{231}\text{Pa}/^{230}\text{Th}$ -data from EW9209 3JPC has been already reported by (Bradtmeier et al., 2014). Why are the values from this paper not included here? Maybe I'm wrong, but I feel there is a lack

of consistency. Or is this maybe just an issue of cutting decimal places? The (Bradtmiller et al., 2014) data would be:

mcd ka Pa/Th

0.06 1.59 0.047

0.76 17.10 0.075

0.96 19.33 0.071

1.06 21.13 0.060

But these values are not visible or slightly different in this manuscript.

Reply #10: EW9209-3JPC data points published in Bradtmiller et al. (2014) were based on an old age model (Curry, 1996). As the core's age model was improved with five ^{14}C dates in this study, the $^{231}\text{Pa}/^{230}\text{Th}$ values were re-calculated to correct for radioactive decay of the actinides using the revised sediment ages. The explanation above is included in the revised manuscript.

5. The authors did not consider $^{231}\text{Pa}/^{230}\text{Th}$ profiles when these show correlations to opal. However, the here applied criteria seem arbitrary to me. I would not disregard time series of $^{231}\text{Pa}/^{230}\text{Th}$ simply because there is a correlation coefficient higher than $r=0.6$ with opal. Why 0.6? And even if so, why should this be a problem? When looking at Supplementary Figure 1a I see a cloud of values clustered between 0 and 0.15 $\text{gcm}^{-2}\text{ k}^{-1}$ and from 0.02 to 0.13 in $^{231}\text{Pa}/^{230}\text{Th}$ with one exception (red crosses core 5). For me this indicates that there is no basin-wide correlation of $^{231}\text{Pa}/^{230}\text{Th}$ with opal as long as opal is not too high (e.g. $\text{gcm}^{-2}\text{ k}^{-1}$), as can be seen in the Southern Ocean. So, what could be a reason for correlations of opal with $^{231}\text{Pa}/^{230}\text{Th}$ for individual cores? I think it is obvious that $^{231}\text{Pa}/^{230}\text{Th}$ increases when a location is bathed by sluggish AABW instead of vigorous NADW. But AABW also hold higher concentrations of silicate and thus sustains higher levels of opal preservation. When neglecting all this locations important paleoceanographic insight will be lost. That's why I do not agree to the statement in line 42, disregarding valuable data sets for interpreting basin wide $^{231}\text{Pa}/^{230}\text{Th}$, which should be revised accordingly.

Reply #11: We agree that overall there is no basin-wide correlation of $^{231}\text{Pa}/^{230}\text{Th}$ with opal in the Atlantic Ocean, as also found in previous time-slice compilation (Bradtmiller et al., 2014; Lippold et al., 2012), as well as in the modern Atlantic (Hayes et al., 2015b). This observation is likely due to the role of AMOC in controlling Atlantic $^{231}\text{Pa}/^{230}\text{Th}$, and potentially the relatively low opal production in the Atlantic. However, ^{231}Pa is more effectively scavenged by opal, an effect shown in experimental studies (Geibert and Usbeck, 2004; Guo et al., 2002; Roberts et al., 2009) and modern observations (Chase et al., 2002; Luo and Ku, 1999; Walter et al., 1997). We feel that we cannot simply dismiss the influence of opal scavenging on past $^{231}\text{Pa}/^{230}\text{Th}$. Including records where there is a significant correlation to opal might add needless complications to the interpretation of past AMOC changes. We have identified $^{231}\text{Pa}/^{230}\text{Th}$ records that are influenced by opal scavenging using the linear correlation method, and the choice of criteria ($r>0.6$) has been explained in detail in the Supplementary Method. Our approach is efficient and straightforward, although we acknowledge that it has limitations. The reviewer is concerned that important paleoceanographic insight will be lost. However, we have made a new Supplementary Figure 2 which shows that including the opal-influenced $^{231}\text{Pa}/^{230}\text{Th}$ records does not change our main interpretations of AMOC: 1) coherent deglacial changes in the western circulation, 2) two-step

AMOC reduction during HS1, 3) some contrasting behaviour between the East and West Atlantic. For our final compilation (Fig. 2), we continue to exclude the opal-influenced $^{231}\text{Pa}/^{230}\text{Th}$ records because: 1) the resulting dataset does not undermine the main paleoceanographic insight presented in this study, 2) we still believe that our approach minimises the potential overprint of (opal) scavenging in sedimentary $^{231}\text{Pa}/^{230}\text{Th}$). The original line 42 is now not appropriate, and has been revised.

6. I would have found it more helpful if the core names would have been given in the figures instead of numbers and latitudes. I'm aware that there is a space issue, but reading would be much easier with core names indicated.

Reply #12: While we agree that many experts would find the core names helpful, we note that for the far broader audience of interested non-experts, the relevant information is not core name, but location and water depth, as we have included in the figures. It is also difficult to keep the figures clear and tidy with the addition of core names. As an alternative, we have added Table 1 to show the list of core names and site locations discussed in the main text. We also keep Supplementary Table 1, which has the full list of sedimentary $^{231}\text{Pa}/^{230}\text{Th}$ time-series examined in this study.

7. The main finding of the difference between east and west circulation during LGM should be elaborated better. There is little evidence from the east (especially when disregarding most of the available profiles there). What is the reason for the very different shape of the profiles from cores GVV01 and RC24-12 compared to SU81-18? And is this number of cores sufficient to allow deriving the east-west gradient in the LGM?

Reply #13: The inferred East-West difference in deep circulation at mid- and low latitudes of North Atlantic during LGM (Fig. 4) and at low latitudes during deglaciation (Fig. 2c) mainly rely on observations from sediment cores from the eastern basin: SU81-18 and GVV01 for the LGM; GVV01 and potentially RC24-12 (note the coarser data resolution in RC24-12) for the deglaciation. Another reviewer, while enthusiastic about the value of the new records from the western basin, also notes the limited data for the East-West comparison. Including the opal-influenced $^{231}\text{Pa}/^{230}\text{Th}$ records (Supplementary Fig. 2) does not significantly contribute to (or detract from) the East-West interpretation. Therefore, we point to the potential for different behaviour of the eastern and western basins, which we also consider intriguing, but highlight the need for future work on more cores from the eastern basin to further examine the East-West contrast, and the different $^{231}\text{Pa}/^{230}\text{Th}$ trends observed at the low latitudes of East Atlantic (GVV01 and RC24-12).

8. line 79: please insert “mainly” between “is” and “derived”, because there are unavoidably contributions from above. These are just smaller, due to the increasing concentrations with water depth.

Reply #14: Revised.

9. line 124: values “[...] range from 0.059 to 124 0.083 and are higher overall than the Holocene (<10 ka) values, which range from 0.041 to 0.065.”. Given these wide ranges within one time period from different site, does it really make sense to create a “composite record”?

Reply #15: Please see reply #7.

10. line 146: although I find the idea of an different AMOC-state at the West and East during the LGM appealing, I wonder if one core (SU81-18) from the North-Atlantic is sufficient to support this hypothesis.

Reply #16: Please see reply #12.

11. line 160: I would rephrase this sentence, because this has been proposed already 13 years before (McManus et al., 2004).

Reply #17: Revised.

12. line 207: I'm not sure what Fig. 4 can tell us here, with most of existing data sets missing. A reduced deep-water transport during the LGM is not new. But how can this be driven by the Labrador Sea Water, which is rather found in the upper NADW?

Reply #18: Reduction of deep-water formation in the Labrador Sea and Irminger Basin (Hillaire-Marcel et al., 2001), shoaling of the other northern-sourced deep waters and expansion of the underlying Antarctic Bottom Water (AABW) in the western basin (Gebbie, 2014) are most likely the combined reasons for reduced western deep-water transport during the LGM. We have included the revised interpretation above in the manuscript.

13. line 240: it would be interesting to calculate how long the cessation of strong NADW formation must be, that it can be recorded by $^{231}\text{Pa}/^{230}\text{Th}$.

Reply #19: The response time of $^{231}\text{Pa}/^{230}\text{Th}$ to changes in circulation strength should be on the scale of ^{231}Pa residence time, which is estimated to be 50–200 years (Henderson and Anderson, 2003). Transient modelling results are consistent with this estimate (Marchal et al., 2000).

14. line 259: "Gulf Stream" is an incorrect oversimplification.

Reply #20: Revised.

15. line 327: Although the authors are well aware of the problems regarding reservoir ages during HS1, they still stick with 400 years. However, the here intrinsic implemented uncertainty endangers any conclusion on trigger and consequence of iceberg (meltwater) purges. The here suggested sequence of events (Eurasian Ice sheets→AMOC reduction→subsurface warming→Laurentide ice sheets→second phase of AMOC reduction), which is contradictive to the scenario proposed by (Barker et al., 2015) is highly dependent on the timing. I wonder if the sequence of events can be maintained with reservoir ages >400 years. Maybe the authors can go through the scenario of higher reservoir ages and test if this conclusion would be still valid.

Reply #21: Age uncertainties form one of the main issues facing sedimentary archives and the comparison of records from different locations. We absolutely agree that surface reservoir uncertainty is one of the biases that needs to be taken into account when considering the phasing of AMOC, meltwater discharge and iceberg calving during the last glacial termination. In the revised manuscript, we have tested the effects of this uncertainty on our conclusions by computing and comparing age models using a range of potential combinations of surface reservoir values expected for high latitudes (>45° N) and low latitudes (<35° N/S) over the last 25 kyr (Stern

and Lisiecki, 2013) (Supplementary Fig. 12). These range from 0–800 years for the low latitudes, and up to 2,000 years for the high latitudes (Stern and Lisiecki, 2013). Overall, despite the potential changes in age, our conclusions on the phasing of AMOC and ice sheet systems during HS1 still hold (see also line 351–353 in the main text).

16. line 354: the certified and measured references values should be given and would be worth a diagram in the supplement.

Reply #22: These are now included (line 366–368).

References from Reviewer 1:

- Barker, S., Chen, J., Gong, X., Jonkers, L., Knorr, G., Thornalley, D., 2015. Icebergs not the trigger for North Atlantic cold events. *Nature* 520, 333-336.
- Bourne, M., Thomas, A., Niocaill, C., Henderson, G., 2012. Improved determination of marine sedimentation rates using ^{230}Th s. *Geochemistry Geophysics Geosystems* 13, Q09017.
- Bradt Miller, L., McManus, J.F., Robinson, L.F., 2014. $^{231}\text{Pa}/^{230}\text{Th}$ evidence for a weakened but persistent Atlantic meridional overturning circulation during Heinrich Stadial 1. *Nature Communications* 5, 5817.
- Burckel, P., Waelbroeck, C., Luo, Y., Roche, D., Pichat, S., Jaccard, S.L., Gherardi, J., Govin, A., Lippold, J., Thil, F., 2016. Changes in the geometry and strength of the Atlantic Meridional Overturning Circulation during the last glacial (20-50 ka). *Climate of the Past* 12, 2061–2075.
- Gherardi, J., Labeyrie, L., Nave, S., Francois, R., McManus, J., Cortijo, E., 2009. Glacial-interglacial circulation changes inferred from $^{231}\text{Pa}/^{230}\text{Th}$ sedimentary record in the North Atlantic region. *Paleoceanography* 24, PA2204.
- Hall, I., Moran, S., Zahn, R., Knutz, P., Shen, C., Edwards, R., 2006. Accelerated drawdown of meridional overturning in the late-glacial Atlantic triggered by transient pre-H event freshwater perturbation. *Geophysical Research Letters* 33, L16616.
- Jonkers, L., Zahn, R., Thomas, A., Henderson, G., Abouchami, W., François, R., Masque, P., Hall, I.R., Bickert, T., 2015. Deep circulation changes in the central South Atlantic during the past 145 kyrs reflected in a combined $^{231}\text{Pa}/^{230}\text{Th}$, Neodymium isotope and benthic record. *Earth and Planetary Science Letters* 419, 14-21.
- Lippold, J., Gutjahr, M., Blaser, P., Christner, E., Ferreira, M.-L.C., Mulitza, S., Christl, M., Wombacher, F., Böhm, E., Antz, B., Cartapanis, O., Vogel, H., Jaccard, S., 2016. Deep water provenance and dynamics of the (de)glacial Atlantic meridional overturning circulation. *Earth and Planetary Science Letters* 445, 68-78.
- Lippold, J., Mulitza, S., Mollenhauer, G., Weyer, S., Christl, M., 2012. Boundary scavenging at the east Atlantic margin does not negate use of Pa/Th to trace Atlantic overturning. *Earth and Planetary Science Letters* 333–334, 317-331.
- Luo, Y., Francois, R., Allen, S., 2010. Sediment $^{231}\text{Pa}/^{230}\text{Th}$ as a recorder of the rate of the Atlantic meridional overturning circulation: insights from a 2-D model. *Ocean Science* 6, 381-400.
- Marchal, O., Francois, R., Stocker, T., Joos, F., 2000. Ocean thermohaline circulation and sedimentary $^{231}\text{Pa}/^{230}\text{Th}$ ratio. *Paleoceanography* 15, 6.
- McManus, J., Francois, R., Gherardi, J., Keigwin, L., Brown-Leger, S., 2004. Collapse and rapid resumption of Atlantic meridional circulation linked to deglacial climate change. *Nature* 428, 834-837.
- Meckler, A.N., Sigman, D.M., Gibson, K.A., Francois, R., Martinez-Garcia, A., Jaccard, S.L., Rohl, U., Peterson, L.C., Tiedemann, R., Haug, G.H., 2013. Deglacial pulses of deep-ocean silicate into the subtropical North Atlantic Ocean. *Nature* 495, 495-498.
- Mulitza, S., Chiessi, C.M., Schefuß, E., Lippold, J., Wichmann, D., Antz, B., Mackensen, A., Paul, A., Prange, M., Rehfeld, K., Werner, M., Bickert, T., Frank, N., Kuhnert, H., Lynch-Stieglitz, J., Portillo-Ramos, R.C., Sawakuchi, A.O., Schulz, M., Schwenk, T., Tiedemann, R., Vahlenkamp, M., Zhang, Y., 2017. Synchronous and proportional deglacial changes in Atlantic meridional overturning and northeast Brazilian precipitation. *Paleoceanography* 32, 622-633.
- Negre, C., Zahn, R., Thomas, A., Masque, P., Henderson, G., Martinez-Mendez, G., Hall, I., Mas, J., 2010. Reversed flow of Atlantic deepwater during the Last Glacial Maximum. *Nature* 468, 84 - 89.
- Roberts, N., McManus, J., Piotrowski, A., McCave, N., 2014. Advection and scavenging controls of Pa/Th in the northern NE Atlantic. *Paleoceanography* 29, 668–679.

Reviewer #2 (Remarks to the Author):

In the manuscript “Coherent deglacial changes in deep Atlantic Ocean circulation”, Ng et al reconstruct AMOC variation in the last deglaciation by sorting 10 out of 23 Pa/Th records and try to provide a basin-wide constraint on the timing and magnitude of the deglacial circulation changes after adjusting chronology. I'm not expert on the proxy, but no surprise to me, even based on a composite dataset the outcome generally falls in line with previously published reconstructions and it seems that the new Pa/Th dataset cannot well capture the abruptness of

AMOC recovery during abrupt BA warming. Is this due to the smoothing choice or the composite approach itself?

Reply #23: The fact that the composite falls in line with prior publications is an important result. To date, the Bermuda Rise record (McManus et al., 2004) has acted as the main reference site for deep Atlantic overturning, and one that is subject to re-interpretation based on inferred local influences (Gil et al., 2009; Lippold et al., 2009). It is important that we have shown a common signal over a wide range of depths and latitudes in the western basin. AMOC recovery at the beginning of the BA is indeed likely to have been more abrupt than recorded in this composite. It is challenging to find sedimentary sites that record the most abrupt features of past changes, and composite has the potential to further reduce this resolution. Furthermore, the response time of $^{231}\text{Pa}/^{230}\text{Th}$ to changes in circulation strength should be on the scale of ^{231}Pa residence time, which is estimated to be 50–200 years (Henderson and Anderson, 2003). However, even with these challenges, we are still able to document rapid changes in the $^{231}\text{Pa}/^{230}\text{Th}$ signal, including a two-step Heinrich Stadial and a YD signal.

In addition, the mechanisms accounting for the AMOC slow-down during the early stage of HS1 and YD are not novel and bit unconvincing. At least, a comprehensive discussion about potential mechanisms, perhaps supported by modeling, is important and necessary for publication in Nature Communications. Based on its current quality, I would suggest at least a major revision before acceptance. Comments below may help improve it.

Reply #24: We have improved the discussion about potential mechanisms of AMOC reduction during HS1 and YD in the revised manuscript (see also reply #24 and #25). Thank you for your comments below, which have helped us with revising the discussion.

1. Regarding triggering mechanisms of AMOC slow-down during deglaciation,

a. the authors try to use a single IDR record from Feni Drift to argue that the early deglaciation of FIS could be the trigger of the first phase of HS1 (~18.5-17ka). But why there is no AMOC slow down during ~21ka BP when there was also a robust IRD release events? See also some similar paper like Peck 2006, EPSL, Toucanne et al 2015 QSR. They also show pre-deglaciation IRD/drainage events, but AMOC is still active at that moment according to Pa/Th. Does the sensitivity of the ocean circulation depend on background climate? This needs to be well discussed.

Reply #25: In the original manuscript, we wrote that “*early AMOC decline was potentially caused by fluvial draining of the Eurasian ice sheets found between 20 and 18 ka*” (original line 185–187). We acknowledge that the choice of an IRD record may not have given the best support to this argument. There are of course other records that indicate early melt from Eurasia, and in the revised manuscript we added the example of a sediment archive of Eurasian fluvial discharge based on the BIT index (Menot et al., 2006) to support the interpretation above (Fig. 3a). Interestingly, the ~21 ka IRD event at Feni Drift coincides with a relatively minor elevation of Eurasian meltwater discharge (Toucanne et al., 2015). If meltwater discharge is the more important driver of AMOC change, then we may not expect a major AMOC decline at ~21 ka.

b. That the authors repeat the ice-sheet melting theory to interpret the YD based on a single IRD record, seems to me that they did not think about this issue seriously. From my point of view, freshwater forcing can be a potential trigger for YD, although the triggering mechanism remains elusive (e.g. Carlson and Clark 2012 Rev Geo; Condrón and Winsor 2012 PNAS). One fundamental

feature is that the sea level rise is just ~ 5m during the YD interval. Given the available evidence from reconstructions and climate modeling, it seems to me that the discrepancy among different hypothesis could be potentially reconciled if the climate system is in a window of AMOC bistability during the YD (e.g. Zhang et al 2014, 2017 NGS). In such a window, a short-lived freshwater perturbation can cause an AMOC slow-down for a long time that can be longer than the duration of the perturbation itself.

Reply #26: We have included a more detailed discussion of YD freshwater forcing in the revised manuscript. We acknowledge that perturbation might have had a longer lasting effect on AMOC when it is in a window of bistability (Zhang et al., 2017). However, given the consistency in the timing of IRD peaks and AMOC decline shown in this study (Fig. 3a & b), we cannot rule out the contribution of melting icebergs in being associated with a reduced AMOC, at least for HS1 and YD.

2. Regarding the two circulation regimes between high latitudes-western Atlantic and eastern Atlantic, the authors employed the GIS ridge overflow to interpret the low Pa/Th value (strong circulation) in the core (34N, 3.14km). If so, why the high-latitude eastern core (50N, 4.28km) that is much closer to the GIS ridge would characterize a high value (weaker circulation)?

Reply #27: There is supporting evidence for a deep northern-sourced water mass in the depth range of 2–3.5 km, between glacial northern-sourced intermediate waters and AABW, in the eastern basin during the LGM (Yu et al., 2008). The $^{231}\text{Pa}/^{230}\text{Th}$ observations at the eastern sites potentially reflect the transition from the deep Nordic water to glacial AABW with depth (Fig. 4a), with low values at the shallowest site (38° N, 3.14 km), and high values at the deepest site (50° N, 4.28 km). We have included the revised interpretation above in the manuscript.

Interestingly, the cores in the eastern side indicate that the deeper the records are the weaker circulation will be. Thus, it is not proper to conclude "a fundamental deglacial reorganization of deep water formation" based on the shallowest record since the rest eastern records indicate a weak circulation during the LGM in comparison to Holocene.

Reply #28: We have revised this section. We would also like to note that the deglacial East-West contrast is based on the low-latitude Sierra Leone Rise record (7° N, 3.43 km) not the shallowest (in the east) Iberian margin record (38° N, 3.14 km).

In general, the core (34N, 3.14km) shows an abnormal feature than the rest, potentially indicating other processes involved, for instance, the NA subtropical circulation. In the North Atlantic, the deep ocean currents are strong in the western (i.e. deep western boundary current) and weak in the eastern. Given the barotropic feature of the subtropical gyre, it is plausible that the record includes a signal from subtropical circulation. According to several modeling studies (e.g. Gong et al 2015 Clim Dyn), it appears that higher northern hemisphere ice sheet can lead to a stronger subtropical circulation. In addition, the Pa/Th value does indicate a weaker circulation during the Holocene than the LGM. Thus, it is plausible that the core (34N, 3.14km) mainly records the strength of the subtropical circulation, rather the deep ocean current. Since several modelers are co-authored in this study, it should be easy to test it by new/published modeling results.

Reply #29: Thank you for the interesting discussion above. We are open to alternative interpretations of the Iberian margin record (38° N, 3.14 km). However, we think that this deep

site is unlikely to be directly influenced by the subtropical circulation at the surface ocean. Sedimentary $^{231}\text{Pa}/^{230}\text{Th}$ is thought to reflect an integrated signal from ~ 1 km of the overlying water column (Thomas et al., 2006). In addition, at deep sites, this $^{231}\text{Pa}/^{230}\text{Th}$ proxy is not sensitive to the signal originating from the surface ocean because the ^{231}Pa and ^{230}Th concentrations at shallow depths are very low (Hayes et al., 2015a; Henderson and Anderson, 2003).

Reviewer #3 (Remarks to the Author):

Ng and co-authors present 4 new Pa/Th reconstructions from the Atlantic Ocean, which cover the last deglaciation. These new cores are added to previously published cores to provide a composite of Pa/Th variations across the deglaciation.

As Pa/Th is a proxy for deep water transport, AMOC changes from the LGM to the Holocene are discussed.

The Pa/Th composite is a welcome addition and nicely confirms previous assumptions about AMOC changes across the deglaciation.

Records are also analyzed with respect to their zonal position in the Atlantic. It is then suggested that deep water transport was a bit stronger in the Eastern part of the Atlantic than in the Western part of the Atlantic at the LGM and until the Holocene.

While I think that the Pa/Th composite is very nice and needed addition, the manuscript could be improved.

Reply #30: We thank the reviewer for their overall positive assessment of our work as a welcome addition and also for specific comments that helped us improve the manuscript.

Please find below some specific comments on the manuscript.

1) The introduction is focused on Pa/Th and challenges associated with it being a proxy for deep water transport. Little to no scientific rationale in generating a Pa/Th composite across the deglaciation is presented. While this would be fine for a more focused journal, I feel that for a journal such as Nature Communication, the introduction should focus on the scientific question to be answered.

Reply #31: On reflection, we agree that the introduction was indeed too focused on the $^{231}\text{Pa}/^{230}\text{Th}$ proxy. Thus, in the revised introduction, we have condensed the paragraphs explaining the $^{231}\text{Pa}/^{230}\text{Th}$ proxy, and have presented clear paragraphs (first and last paragraphs of introduction) explaining the scientific question to be answered and the rationale of this study (including the generation of the $^{231}\text{Pa}/^{230}\text{Th}$ composite).

2) The “Results” section is extremely short and in fact just gives a very broad overview of Pa/Th changes across the deglaciation: i) overview of Pa/Th values at different times of the deglaciation

without any specific context, ii) two phases changes during HS1, iii) East-West contrast. Please note that p7, L. 145-146 it is noted that the Rockall plateau Pa/Th values are similar at the LGM and Holocene, which does not seem to be the case in figs 2 and 4...

Reply #32: We think the results are self-contained, with clear subheadings containing comprehensive description of data. The exciting interpretations and implications are in the following discussion section. In the original manuscript we wrote “*higher LGM $^{231}\text{Pa}/^{230}\text{Th}$ in the Equatorial Atlantic (3°S , 3.49 km, Fig. 2c & 4) and Rockall Basin (50°N , 4.28 km, Fig. 2a & 4)*”, and we believe this accurately summarizes those data. The immediately preceding line included the phrase “similar $^{231}\text{Pa}/^{230}\text{Th}$ at Sierra Leone Rise”, which may have led to the potential confusion above.

3) The discussion starts with HS1, which is very interesting and most likely the most interesting part of the ms as also highlighted in the abstract. But then it goes onto discussing the East-West gradient across the last 25 kyrs. This different behavior zonally across the Atlantic is not as robust as the rest and, if anything, is mostly seen at the LGM. As such most of the discussion focuses on the LGM. This gives rise to 2 issues:

i) The discussion starts with HS1, moves to the LGM and then fairly quickly across the deglaciation with most events mentioned briefly in one paragraph. This break in the chronology is a bit distracting and all the succession of BA, YD, Holocene reads like a list with approximation.

Reply #33: As suggested, we have restructured the discussion section and now it starts with examining the two-phase AMOC decline from LGM to HS1, followed by discussion of the late deglaciation: BA and YD. Then, we note the potential East-West difference during LGM and deglaciation. We have also added discussion on YD following the suggestions of Reviewer 2.

ii) The zonal difference relies mostly on one previously published core: SU81-18 from Gherardi et al., 2005.

The data from the Iberian margin and the Sierra Leone Rise lend support to the zonal difference at the LGM. However, I'm not sure the evidence for zonal differences during the deglaciation is robust. The Iberian margin core nicely displays H1, BA, YD variations, which seem to follow the composite pa/Th very closely: i.e. I'm not sure the East-West difference is statically significant outside of the LGM.

Cores 7 and 10 are somewhat lower resolution, their variations in general more muted. These cores are “quite far” from the NADW path, which flows as a western boundary current in the deep Atlantic. These cores could be influenced by changes in the upwelling system instead.

Reply #34: Reviewer 1 has also raised their interests and concerns on the number of (eastern) records to constrain the East-West difference, please see reply #12. We welcome alternative interpretations on the Sierra Leone Rise core (7°N , 3.43 km, originally core 7, now core 8) and the Equatorial Atlantic core (3°S , 3.49 km, originally core 10, now core 12). If the $^{231}\text{Pa}/^{230}\text{Th}$ records from these deep low-latitude eastern sites are influenced by changes in the upwelling system, we would expect them to mirror $^{231}\text{Pa}/^{230}\text{Th}$ variations in the shallower African margin core (15°N , 2.38 km) which lies on a coastal upwelling site (Supplementary Fig. 1). The African margin core records a substantial increase of $^{231}\text{Pa}/^{230}\text{Th}$ over the last deglaciation (Supplementary Fig. 2), but the deep low-latitude eastern sites show no evident $^{231}\text{Pa}/^{230}\text{Th}$ changes during this period (Fig. 2c).

Therefore, given available evidence, we think that the Sierra Leone Rise core and the Equatorial Atlantic core are unlikely to be influenced by the upwelling system.

Figures: The figures are very nice and informative.

Reply #35: We appreciate this positive assessment, as we have put some care and effort into their construction.

Figure 1:

An useful system of symbols and numbers is set up to identify cores. This is very nice, but core names and references should at least be added here somewhere either on the figure or in the legend. Else the supplementary table should be moved to the main text.

Reply #36: We agree with the first point, and have also taken the second to heart, and thus added this information. Please see reply #11.

Minor points:

-p2, L. 38: “relies”

-p7, L. 147: no coordinates are given for the Sierra Leone Rise core

-p9, L. 195: maybe “mid HS1” might be more appropriate than “late HS1”.

-p10, L. 230; “These rate proxy records...” needs rephrasing

p12, L. 262: “This data supports”

- P12, L.266-267: I am not sure “an in-depth analysis of the LGM-deglacial phasing of ocean circulation, ice-sheet and climate changes” was given in the ms.

Reply #37: We have made revisions following the comments above, except for the third point – we think that late HS1 is appropriate, as we are referring to the fact that the reduction was sustained throughout late HS1 until its end, rather than the fact that the second phase of reduction began in mid-HS1.

Authors' references:

Bourne, M.D., Thomas, A.L., Mac Niocaill, C., Henderson, G.M., 2012. Improved determination of marine sedimentation rates using Th-230(xs). *Geochem Geophys Geosy* 13.

Bradt Miller, L.I., McManus, J.F., Robinson, L.F., 2014. 231Pa/230Th evidence for a weakened but persistent Atlantic meridional overturning circulation during Heinrich Stadial 1. *Nat Commun* 5.

Chase, Z., Anderson, R.F., Fleisher, M.Q., Kubik, P.W., 2002. The influence of particle composition and particle flux on scavenging of Th, Pa and Be in the ocean. *Earth Planet Sc Lett* 204, 215-229.

Curry, W.B., 1996. Late Quaternary deep circulation in the western equatorial Atlantic.

Gebbie, G., 2014. How much did Glacial North Atlantic Water shoal? *Paleoceanography* 29, 190-209.

Geibert, W., Usbeck, R., 2004. Adsorption of thorium and protactinium onto different particle types: Experimental findings. *Geochim Cosmochim Acta* 68, 1489-1501.

Gil, I.M., Keigwin, L.D., Abrantes, F.G., 2009. Deglacial diatom productivity and surface ocean properties over the Bermuda Rise, northeast Sargasso Sea. *Paleoceanography* 24, PA4101.

Guo, L.D., Chen, M., Gueguen, C., 2002. Control of Pa/Th ratio by particulate chemical composition in the ocean. *Geophys Res Lett* 29.

Hayes, C.T., Anderson, R.F., Fleisher, M.Q., Huang, K.F., Robinson, L.F., Lu, Y.B., Cheng, H., Edwards, R.L., Moran, S.B., 2015a. Th-230 and Pa-231 on GEOTRACES GA03, the US GEOTRACES North Atlantic transect, and implications for modern and paleoceanographic chemical fluxes. *Deep-Sea Res Pt II* 116, 29-41.

Hayes, C.T., Anderson, R.F., Fleisher, M.Q., Vivanco, S.M., Lam, P.J., Ohnemus, D.C., Huang, K.F., Robinson, L.F., Lu, Y.B., Cheng, H., Edwards, R.L., Moran, S.B., 2015b. Intensity of Th and Pa scavenging partitioned by particle chemistry in the North Atlantic Ocean. *Mar Chem* 170, 49-60.

Henderson, G.M., Anderson, R.F., 2003. The U-series toolbox for paleoceanography. *Rev Mineral Geochem* 52, 493-531.

Hillaire-Marcel, C., de Vernal, A., Bilodeau, G., Weaver, A.J., 2001. Absence of deep-water formation in the Labrador Sea during the last interglacial period. *Nature* 410, 1073-1077.

Lippold, J., Grutzner, J., Winter, D., Lahaye, Y., Mangini, A., Christl, M., 2009. Does sedimentary Pa-231/Th-230 from the Bermuda Rise monitor past Atlantic Meridional Overturning Circulation? *Geophys Res Lett* 36, L12601.

Lippold, J., Luo, Y.M., Francois, R., Allen, S.E., Gherardi, J., Pichat, S., Hickey, B., Schulz, H., 2012. Strength and geometry of the glacial Atlantic Meridional Overturning Circulation. *Nat Geosci* 5, 813-816.

Luo, S.D., Ku, T.L., 1999. Oceanic Pa-231/Th-230 ratio influenced by particle composition and remineralization. *Earth Planet Sc Lett* 167, 183-195.

Marchal, O., Francois, R., Stocker, T.F., Joos, F., 2000. Ocean thermohaline circulation and sedimentary Pa-231/Th-230 ratio. *Paleoceanography* 15, 625-641.

McManus, J.F., Francois, R., Gherardi, J.M., Keigwin, L.D., Brown-Leger, S., 2004. Collapse and rapid resumption of Atlantic meridional circulation linked to deglacial climate changes. *Nature* 428, 834-837.

Menot, G., Bard, E., Rostek, F., Weijers, J.W.H., Hopmans, E.C., Schouten, S., Damste, J.S.S., 2006. Early reactivation of European rivers during the last deglaciation. *Science* 313, 1623-1625.

Roberts, K.A., Xu, C., Hung, C.C., Conte, M.H., Santschi, P.H., 2009. Scavenging and fractionation of thorium vs. protactinium in the ocean, as determined from particle-water partitioning experiments with sediment trap material from the Gulf of Mexico and Sargasso Sea. *Earth Planet Sc Lett* 286, 131-138.

Stern, J.V., Lisiecki, L.E., 2013. North Atlantic circulation and reservoir age changes over the past 41,000 years. *Geophys Res Lett* 40, 3693-3697.

Thomas, A.L., Henderson, G.M., Robinson, L.F., 2006. Interpretation of the Pa-231/Th-230 paleo circulation proxy: New water-column measurements from the southwest Indian Ocean. *Earth Planet Sc Lett* 241, 493-504.

Toucanne, S., Soulet, G., Freslon, N., Silva Jacinto, R., Dennielou, B., Zaragosi, S., Eynaud, F., Bourillet, J.-F., Bayon, G., 2015. Millennial-scale fluctuations of the European Ice Sheet at the end of the last glacial, and their potential impact on global climate. *Quaternary Sci Rev* 123, 113-133.

Voigt, I., Cruz, A.P.S., Mulitza, S., Chiessi, C.M., Mackensen, A., Lippold, J., Antz, B., Zabel, M., Zhang, Y., Barbosa, C.F., Tisserand, A.A., 2017. Variability in mid-depth ventilation of the western Atlantic Ocean during the last deglaciation. *Paleoceanography* 32, 948-965.

Walter, H.J., vanderLoeff, M.M.R., Hoeltzen, H., 1997. Enhanced scavenging of Pa-231 relative to Th-230 in the south Atlantic south of the Polar front: Implications for the use of the Pa-231/Th-230 ratio as a paleoproductivity proxy. *Earth Planet Sc Lett* 149, 85-100.

Yu, J.M., Elderfield, H., Piotrowski, A.M., 2008. Seawater carbonate ion-delta C-13 systematics and application to glacial-interglacial North Atlantic ocean circulation. *Earth Planet Sc Lett* 271, 209-220.

Zhang, X., Knorr, G., Lohmann, G., Barker, S., 2017. Abrupt North Atlantic circulation changes in response to gradual CO2 forcing in a glacial climate state. *Nat Geosci* 10, 518.

Reviewers' comments:

Reviewer #1 (Remarks to the Author):

This is my second review of this manuscript. I remain convinced that the new $^{231}\text{Pa}/^{230}\text{Th}$ results presented are interesting and that valuable lessons can be learned from it. However, the revision of the manuscript did not settle all of my concerns. Here I respond on the replies by the authors. I will skip all issues when these points are satisfyingly settled, but I keep the continuing numbers for the others. This is followed by a few new points.

Reply #7: The disequilibria that may result from alpha recoil is not explicitly taken into account in our calculations, as we do not have an independent measure of this effect, but there is negligible difference in correcting the $^{231}\text{Pa}/^{230}\text{Th}$ data for alpha recoil by assuming lithogenic $^{234}\text{U}/^{238}\text{U}$, $^{230}\text{Th}/^{234}\text{U}$, $^{231}\text{Pa}/^{235}\text{U}$ of 0.96 (Bourne et al., 2012) instead of 1 (secular equilibrium).

There is a potential disequilibrium for $^{234}\text{U}/^{238}\text{U}$, only. There is no need to apply 0.96 on the $^{231}\text{Pa}/^{235}\text{U}$, of course. I assume that this correction suggested by (Bourne et al., 2012) may not have a big effect on these values. However, I would appreciate if neglecting this correction would at least be mentioned in the method part.

Reply #8: We agree that $^{231}\text{Pa}/^{230}\text{Th}$ changes with latitude and water depth. However, the purpose of our composite record is not to compute a single absolute $^{231}\text{Pa}/^{230}\text{Th}$ value for the whole (deep North) Atlantic at any single point of time (which would be an impossible task), but to show the average trends and amplitudes. Despite the depth and latitudinal variations, all records from the west and deep high latitude North Atlantic still display coherent $^{231}\text{Pa}/^{230}\text{Th}$ trends over the last 25 kyr (Fig. 2a), which suggests that our approach of constructing a composite record from these cores to reflect overall large-scale changes in AMOC strength is appropriate. Converting this composite record to an overall 'rate' or water mass transport would require more detailed modelling. For the above reasons, we decide to keep the composite record in the revised manuscript, as well as showing the $^{231}\text{Pa}/^{230}\text{Th}$ data from individual records that were used to construct the composite (Fig. 3b).

This issue has not been solved and still needs revision. The authors say that they do not intend to compute a single absolute $^{231}\text{Pa}/^{230}\text{Th}$ value. But this is what they are doing here. What is the advantage of an average $^{231}\text{Pa}/^{230}\text{Th}$ value between a location at 50° close to the deep water formation zone and one in the subtropical intermediate Ocean all the way down and 2km shallower? It is not appropriate to simply average these values. Averaged amplitudes do not make much sense either, because the range of values is a function of depth. As the authors correctly point out in Reply #29 sedimentary $^{231}\text{Pa}/^{230}\text{Th}$ reflects an integrated signal from ~ 1 km of the overlying water column making absolute values very sensitive on water depth. If the authors are really interested in trends (or rather variability) they need to do some serious time series analysis. Not only a 9-point-moving average (why 9?). Including normalisation, error analysis and weighting, consideration of time resolution, temporal signal distribution, binning etc. But even if they would do so, what information does this contain? Let's imagine another parameter instead of $^{231}\text{Pa}/^{230}\text{Th}$, let's say oxygen concentration, which might be more catchy than this conceptual measured variable of $^{231}\text{Pa}/^{230}\text{Th}$. Would it be appropriate to apply a 9-point-moving average on oxygen concentrations from several locations starting at 50° N directly affected by ventilation and deep water formation reaching towards one in the subtropical intermediate Ocean with very different particle fluxes and a very different history all the way down? I don't think so. This would only make sense, when restricted to a very distinct and constrained water mass (not covering 2 km of water depth and almost half of the Atlantic in latitude), or when homogeneously distributed all over the Atlantic, which would be a copy of the approach by Yu et al. back in 1996, who wasn't aware of the water-depth-effect.

Summarized, in this oversimplified form such a mixed $^{231}\text{Pa}/^{230}\text{Th}$ -record cannot be accepted in a publication in Nature Communications.

Reply #9: We made every effort to include all available cores – and some were initially excluded in the original manuscript as they were at shallower intermediate depths. However, in response to the reviewer's comment we have decided to include all intermediate-depth cores as well – as can be seen in the supplements. In the revised manuscript, we have added MD95-2037, MD02-2594, IODP1313, GeoB16202-2, DAPC2, BOFS 17K, GeoB9508-1, and the more recent GeoB16206-1 (Voigt et al., 2017) $^{231}\text{Pa}/^{230}\text{Th}$ profiles. The following cores have not been included for the following reasons:

- a) MD03-2705: there is ^{230}Th data but not ^{231}Pa data.
- b) MD02-2588: it is in the Indian Ocean.
- c) GeoB3808-6: the data resolution is too low to be used in this study – there are only five data points over 25–0 ka.

Here the authors made a way better job providing now a thorough overview of the available data. I agree to neglect MD02-2588 and GeoB3808-6 due to the reasons given by the authors. They could still think about including MD03-2705. Its available via <https://doi.pangaea.de/10.1594/PANGAEA.810309> because they are right, the ^{231}Pa is curiously not listed in their supplement of Meckler et al.

Reply #10: EW9209-3JPC data points published in Bradtmiller et al. (2014) were based on an old age model (Curry, 1996). As the core's age model was improved with five ^{14}C dates in this study, the $^{231}\text{Pa}/^{230}\text{Th}$ values were re-calculated to correct for radioactive decay of the actinides using the revised sediment ages. The explanation above is included in the revised manuscript.

I welcome this improvement. Please indicate this clearly in the supplement (caption S8) as well.

Reply #11: We agree that overall there is no basin-wide correlation of $^{231}\text{Pa}/^{230}\text{Th}$ with opal in the Atlantic Ocean, as also found in previous time-slice compilation (Bradtmiller et al., 2014; Lippold et al., 2012), as well as in the modern Atlantic (Hayes et al., 2015b). This observation is likely due to the role of AMOC in controlling Atlantic $^{231}\text{Pa}/^{230}\text{Th}$, and potentially the relatively low opal production in the Atlantic. However, ^{231}Pa is more effectively scavenged by opal, an effect shown in experimental studies (Geibert and Usbeck, 2004; Guo et al., 2002; Roberts et al., 2009) and modern observations (Chase et al., 2002; Luo and Ku, 1999; Walter et al., 1997). We feel that we cannot simply dismiss the influence of opal scavenging on past $^{231}\text{Pa}/^{230}\text{Th}$. Including records where there is a significant correlation to opal might add needless complications to the interpretation of past AMOC changes. We have identified $^{231}\text{Pa}/^{230}\text{Th}$ records that are influenced by opal scavenging using the linear correlation method, and the choice of criteria ($r > 0.6$) has been explained in detail in the Supplementary Method. Our approach is efficient and straightforward, although we acknowledge that it has limitations. The reviewer is concerned that important paleoceanographic insight will be lost. However, we have made a new Supplementary Figure 2 which shows that including the opal-influenced $^{231}\text{Pa}/^{230}\text{Th}$ records does not change our main interpretations of AMOC: 1) coherent deglacial changes in the western circulation, 2) two-step AMOC reduction during HS1, 3) some contrasting behaviour between the East and West Atlantic. For our final compilation (Fig. 2), we continue to exclude the opal-influenced $^{231}\text{Pa}/^{230}\text{Th}$ records because: 1) the resulting dataset does not undermine the main paleoceanographic insight presented in this study, 2) we still believe that our approach minimises the potential overprint of (opal) scavenging in sedimentary $^{231}\text{Pa}/^{230}\text{Th}$. The original line 42 is now not appropriate, and has been revised.

I still consider the here applied criterion as somehow arbitrary and selective. As the authors point

out with the new Supplementary Figure 2, when including $^{231}\text{Pa}/^{230}\text{Th}$ records which are influenced by opal in their opinion, this not change the main interpretation. This gives rise to the assumption that the criterion is not full developed.

When a linear correlation of 0.6 is a general applicable criterion, I wonder why one locations falls within this selection, while a very adjacent location doesn't, given that Pa/Th records large scale effects. This is the case for e.g.: [25] and [31], [11] and [24], [1] and [14]. However, it is up to them if they unnecessarily aim for reducing the available information.

New comments:

e.g. line 125: I cannot see a "gradual increase". Given the error bars this is a steady peak rather than an increase. Please consider rephrasing.

e.g. line 131: Here it should be considered and mentioned that the eastern cores are all from a very similar water depth, resulting in very similar $^{231}\text{Pa}/^{230}\text{Th}$ profiles.

It should be explained and elaborated what goes on during YD at core GVV14.

Core IODP U1313: Why is this core considered as an eastern core (Fig. 4)?

Reviewer #2 (Remarks to the Author):

The revised version by Ng et al provides several insights about the composite $^{231}\text{Pa}/^{230}\text{Th}$ observations during the last deglaciation. According to the composite, it is easy to identify phases of deglacial ocean circulation changes. But I'm a bit doubtful whether it can well represent the exact timing when the system went across tipping point if it exists since the signal was smoothed, although I understand that it can provide an overall estimate of past circulation changes. Therefore, caution should be paid to the exact timing of abrupt AMOC shifts.

Regarding my previous comments, I like the authors' idea to interpret the $^{231}\text{Pa}/^{230}\text{Th}$ records at the eastern sites as the indication of a water-mass transition from the deep Nordic to the Southern Ocean. However, the dynamics regarding the occurrence of YD remains unclear. Other comments are listed in the following:

Abstract: please specify the timing of different steps of deglacial AMOC changes. Otherwise, it is not clear about the timing of the events during the deglaciation for readers.

Line 24 "reduced" => "weakening"

Line 37:

add literature Zhang et al (2017 NGS) after "carbon budget", and Zhang et al (2014 Nature) after "ice sheet volume", which are relevant and supportive here since they are to explore the AMOC responses to atmospheric CO₂ and ice volume.

Line 54-56:

Here the authors argue that $^{231}\text{Pa}/^{230}\text{Th}$ is proposed to reflect the integrated signal of the overlying ~1km water column when there is active water-mass advection. Then could I consider the $^{231}\text{Pa}/^{230}\text{Th}$ in the deep North Atlantic (lower than ~3.2 km of the water depth) represents the strength of the AABW cell of the AMOC during the LGM, given that the deep Atlantic was bathed by the AABW then and the boundary between GNAIW and AABW ($\delta^{13}\text{C}$ 0.8 isoline is at the water depth of ~ 2.2km in the North Atlantic (Curry and Oppo 2005))? If so, what's the implication for the records in the western sites in the northern North Atlantic?

Line 124-125: "a distinct early rise from 18.5 ka to 17 ka, followed by ~2,000 years of a gradual increase to peak values from 17 to 15 ka (Fig. 3b)."

What're the criteria to define the boundary between the two phases of HS1? From my point of view, the boundary should be around 16.2ka BP. Several reasons (not all) are listed in the following: 1) Before that, the composite 231Pa/230Th increased gradually (Big Dry episode), followed by an abrupt rise at around 16.2ka BP associated with a shut-down AMOC (Big Wet episode). 2) melting water contribution of the Eurasian ice sheet is trivial after 16.2ka BP, indicating a major role of LIS on the second phase of HS1. 3) Of the 231Pa/230Th at the Iberian margin the gradual rise ends at around 16.2ka BP. 4) rate of increases in atmospheric CO₂ is evident in the first phase, followed by a CO₂ plateau in the second phase.

Line 175-178: 'The observed ... freshwater forcing':

Visually, 231Pa/230Th slightly decreases from 20ka to 19ka BP, indicating a strengthening AMOC that can lead to a negative mass balance of Eurasian Ice sheet and an increase in the fluvial drainage. Once the freshwater input hits the peak at around 19ka, the AMOC starts to decrease, followed by subsurface warming and IRD release from Eurasian Ice Sheet.

Line 178-180: 'Reduced AMOC ... (Fig. 3a)'

The reference is not quite proper since it is specific to ice-shelves from Laurentide Ice Sheet. Additionally, it is not clear to me why the marine-based Eurasian ice shelves are more sensitive to the subsurface warming than the Laurentide ice sheet as the AMOC slows down. Probably, it is associated with the location of the freshwater input (NE North Atlantic), which is more effective on modulating the vertical structure of Nordic sea than the Labrador Sea. The authors need to discuss a bit on this issue.

Line 188-190: 'This AMOC ... period'

The defined boundary between the two phases does not fits the timing of REF37. Please refer to my comments regarding the definition.

Line 198-202: 'The timing of AMOC ... ice sheets'

This argument in this sentence is a bit strong. The IRD records just represent the calving part of the ice sheets, and the BIT index only associated with meltwater from the Eurasian ice sheet. All of them cannot rule out the existence of persistent meltwater from Laurentide ice sheet during BA. Furthermore, within the age uncertainty, there is a rapid sea level rise at the same time, which indicates that BA warming is accompanied with freshwater input, in contrast to the argument here. One alternative explanation for BA warming associated with AMOC resumption is due to the increased atmospheric CO₂ (Zhang et al 2017 NGS), since the increasing CO₂ can overcome the effect of freshwater input on the AMOC strength.

Line 211-214: 'instead ... sea-ice model'

"most likely" is a bit strong here. As I pointed out in last time, it is not that simple to provide a reasonable triggering mechanism for YD (please see a potential new triggering mechanism of YD published in CPD recently -- Baldini et al 2017 CPD). The IRD record from New Found land indeed indicates a contemporary melting iceberg that sustains the cold condition. However, the amount of meltwater input from Icebergs could be smaller and shorter than that during HS1 (e.g. ~0.09Sv for 300 years during YD [Tarasov and Peltier 2005]), which is hard to explain the 1200-year-long YD event by coupled GCMs. Evidence regarding the timing of IRD events also indicate that iceberg discharge occurred after the start of the YD (Hillaire-Marcel and Bilodeau, 2000). The concept of AMOC stability can potentially overcome this problem. I'm a bit disappointed as the AMOC stability is not mentioned in this context. From my point of view, YD is a good testbed to evaluate its linkage to AMOC stability associated with ice volume changes (Zhang et al 2014), as the global mean sea level is just at its intermediate level during the glacial-interglacial cycles. This indicates that the climate system under the YD background condition is more sensitive to forcing that can

modulate the strength of the AMOC. Therefore, even a temporal forcing under this context can lead to a long-term persistent consequence on the climate system.

Line 235:

'ref. 51': is not proper here. In the abstract of REF 51, it proposed that the interface shoals but the vertical volume of NADW are not different from modern day, which is a contrast to "weakened" as authors stated here.

Line 236:

'ref. 47' is not proper here. (This paper is to discuss interglacial rather glacial climate)

Line 249-250:

Nice point here. So, is it plausible that ocean circulation in the upper level (above 3km) can be stronger during the LGM than the modern day in the North Atlantic? what's its implication for the western cores?

Reviewer #3 (Remarks to the Author):

2nd review of Ng et al.

The authors have taken into account some of the comments that were made during the 1st review. I think the manuscript has improved and reads better than in the first version. Overall, I think it is a welcome addition and a nice paper, for which I recommend publication after minor revisions. However, I should point out that I am still unconvinced by some of the interpretation made, particularly with respect to the East-West contrast.

1) The authors suggest that deep-water circulation was stronger in the Northeast Atlantic than in the Northwest. They further suggest this could be due to changes in the location of deep water formation. However, even if deep water was only formed in the Nordic Sea, it would still flow into the western part of the Atlantic as a deep western boundary current and a minor part of the flow would go into the Eastern part of the basin. As detailed below, there is little evidence to suggest that the eastern equatorial Atlantic cores record northern sourced deep water masses. Therefore the East-West discrepancy mostly relies on the Iberian margin core (and a few points from MAR).

2) I am not convinced about interpretation of cores 8 and 12 (GVY01 and RC24). The relationship between Pa/Th has been tested in the Northwest Atlantic (Rempfer et al., 2017), but I am not sure of the AMOC influence on the eastern equatorial Atlantic. Assuming there is a deep Northeast water mass flowing south, then it should be weaker at GVV01 than at the Iberian Margin. But the relationship between changes in these cores and the Iberian margin is weak. In addition, this core could also be under the influence of Antarctic Bottom Water (AABW), but AABW would first influence the Brazilian margin (cores 9 and 10), before these ones. Therefore, contrarily to what expected these 2 cores show the lowest Pa/Th values across the deglaciation and the smallest changes. This means that these 2 cores most likely reflect something else than North Atlantic deep water masses or AABW.

3) Two phases of HS1: the authors distinguish 2 phases for HS1, an early and late HS1. If the authors really want to define phases for HS1, then maybe a bit more thought and/or justification should be put into that effort.

i) The authors never actually give time-frames for these 2 periods (e.g. from 18.6 to 17ka and from 17 to 15.4ka), which are only seen in Figure 3.

ii) These two phases are separated by the increase in BIT index (at 17ka) and more or less coincide with the Pa/Th record. The early phase broadly refers to the time period where Pa/Th increases and where some IRD peaks are seen in the Rockall Basin and fluvial discharges in the

Bay of Biscay. But the composite Pa/Th reaches almost maximal values at 17.5 ka, meaning that the AMOC is most likely very weak starting from that point on, with important implications for the climate and carbon cycle. Shouldn't the early phase stop at 17.5ka?

iii) There is another slight increase in Pa/Th at 16ka. The authors mention this could be coincident with observations of changes in the hydrological cycle. But I wonder, are these changes in Pa/Th really significant? Do the authors really think that this change at 16ka really represent a change in the formation and transport of deep-water mass?

iv) The recovery from HS1 (15.5-14.5 ka) is not labeled as anything.... Maybe late HS1 should be extended to ~14.7 ka, when NGRIP d18O increases abruptly.

Response to reviewers

Key:

- Reviewers' comments
- Author's response

Reviewer #1 (Remarks to the Author):

This is my second review of this manuscript. I remain convinced that the new $^{231}\text{Pa}/^{230}\text{Th}$ results presented are interesting and that valuable lessons can be learned from it. However, the revision of the manuscript did not settle all of my concerns. Here I respond on the replies by the authors. I will skip all issues when these points are satisfyingly settled, but I keep the continuing numbers for the others. This is followed by a few new points.

Response #1: We thank the reviewer for their continued interest in our new $^{231}\text{Pa}/^{230}\text{Th}$ results. Below we have addressed their comments to improve the manuscript.

Reply #7: The disequilibria that may result from alpha recoil is not explicitly taken into account in our calculations, as we do not have an independent measure of this effect, but there is negligible difference in correcting the $^{231}\text{Pa}/^{230}\text{Th}$ data for alpha recoil by assuming lithogenic $^{234}\text{U}/^{238}\text{U}$, $^{230}\text{Th}/^{234}\text{U}$, $^{231}\text{Pa}/^{235}\text{U}$ of 0.96 (Bourne et al., 2012) instead of 1 (secular equilibrium).

There is a potential disequilibrium for $^{234}\text{U}/^{238}\text{U}$, only. There is no need to apply 0.96 on the $^{231}\text{Pa}/^{235}\text{U}$, of course. I assume that this correction suggested by (Bourne et al., 2012) may not have a big effect on these values. However, I would appreciate if neglecting this correction would at least be mentioned in the method part.

Response #2: Revised.

Reply #8: We agree that $^{231}\text{Pa}/^{230}\text{Th}$ changes with latitude and water depth. However, the purpose of our composite record is not to compute a single absolute $^{231}\text{Pa}/^{230}\text{Th}$ value for the whole (deep North) Atlantic at any single point of time (which would be an impossible task), but to show the average trends and amplitudes. Despite the depth and latitudinal variations, all records from the west and deep high latitude North Atlantic still display coherent $^{231}\text{Pa}/^{230}\text{Th}$ trends over the last 25 kyr (Fig. 2a), which suggests that our approach of constructing a composite record from these cores to reflect overall large-scale changes in AMOC strength is appropriate.

Converting this composite record to an overall 'rate' or water mass transport would require more detailed modelling. For the above reasons, we decide to keep the composite record in the revised manuscript, as well as showing the $^{231}\text{Pa}/^{230}\text{Th}$ data from individual records that were used to construct the composite (Fig. 3b).

This issue has not been solved and still needs revision. The authors say that they do not intend to compute a single absolute $^{231}\text{Pa}/^{230}\text{Th}$ value. But this is what they are doing here. What is the advantage of an average $^{231}\text{Pa}/^{230}\text{Th}$ value between a location at 50° close to the deep water formation zone and one in the subtropical intermediate Ocean all the way down and 2km shallower?

Response #3: The sediment core records used to develop the composite curve are from locations proximal to deep water formation sites (50° N, 4.28 km), and locations that are relatively responsive to changes in AMOC strength (Lippold et al., 2011; Rempfer et al., 2017): Brazil margin (2° S, 2.25 km) and the deep western basin of North Atlantic (34° N, 4.55 km; 15° N, 2.71 km; 6° N, 4.06 km; 5° N, 3.29 km). In our compilation, we have shown that these cores exhibit coherent, large changes in $^{231}\text{Pa}/^{230}\text{Th}$ over the last 25 ka that are driven by changes in AMOC strength, despite the large range of water depths, latitudes, and oceanic environments the cores sit at. We have taken the reviewer and editor comments on board, and removed the composite from the main text. For reference to readers who do want to see what a composite would look like we have moved it to Supplementary Fig. 6, including a description of the composite's limitations in the supplementary figure caption.

It is not appropriate to simply average these values. Averaged amplitudes do not make much sense either, because the range of values is a function of depth. As the authors correctly point out in Reply #29 sedimentary $^{231}\text{Pa}/^{230}\text{Th}$ reflects an integrated signal from ~ 1 km of the overlying water column making absolute values very sensitive on water depth. If the authors are really interested in trends (or rather variability) they need to do some serious time series analysis. Not only a 9-point-moving average (why 9?). Including normalisation, error analysis and weighting, consideration of time resolution, temporal signal distribution, binning etc.

Response #4: It would be great to have enough data at a suitable resolution for full time-series analyses, but that is not yet possible. Our choice of 9-point moving average indeed does have limitations – but it is suited to the data resolution, because it integrates the $^{231}\text{Pa}/^{230}\text{Th}$ data over 500–1,000 years, which is a suitable compromise for accounting sediment chronology uncertainty and preserving changes on the millennial timescale. We have moved the composite to the supplements to place the emphasis on the individual records, and we have also derived an alternative composite curve using another statistical method (Supplementary Fig. 7) to address the reviewer's concerns. The method involves binning the $^{231}\text{Pa}/^{230}\text{Th}$ data at equally spaced 500-year intervals from 20–10 ka (and binning data at equally spaced 1,000-year intervals from 10–1 ka and 24–20 ka due to lower data resolution). The 1 standard deviation of the binned dataset then represents the range of $^{231}\text{Pa}/^{230}\text{Th}$ data given the range of water depths, latitudes, and oceanic environment the cores sit at (Supplementary Fig. 7), although we note that the individual data points plotted with the original composite curve (Supplementary Fig. 6) also represent the range of $^{231}\text{Pa}/^{230}\text{Th}$ from the different core sites. More importantly, this range of $^{231}\text{Pa}/^{230}\text{Th}$ data does not undermine the coherent millennial-scale signal observed in the west and deep high-latitude North Atlantic (Supplementary Fig. 6 & 7). Notably, the alternative composite curve developed using the data binning method (Supplementary Fig. 7) is very similar to the original one developed using the moving average method (Supplementary Fig. 6).

But even if they would do so, what information does this contain? Let's imagine another parameter instead of $^{231}\text{Pa}/^{230}\text{Th}$, let's say oxygen concentration, which might be more catchy than this conceptual measured variable of $^{231}\text{Pa}/^{230}\text{Th}$. Would it be appropriate to apply a 9-point-moving average on oxygen concentrations from several locations starting at 50°N directly affected by ventilation and deep water formation reaching towards one in the subtropical intermediate Ocean with very different particle fluxes and a very different history all the way down? I don't think so. This would only make sense, when restricted to a very distinct and constrained water mass (not covering 2 km of water depth and almost half of the Atlantic in latitude), or when homogeneously distributed all over the Atlantic, which would be a copy of the approach by Yu et al. back in 1996, who wasn't aware of the water-depth-effect. Summarized, in this oversimplified form such a mixed $^{231}\text{Pa}/^{230}\text{Th}$ -record cannot be accepted in a publication in Nature Communications.

Response #5: The composite approach has been removed from the main text to allay the reviewer's concerns, and a more detailed discussion of the limitations is given in the supplements.

Reply #9: We made every effort to include all available cores – and some were initially excluded in the original manuscript as they were at shallower intermediate depths. However, in response to the reviewer's comment we have decided to include all intermediate-depth cores as well – as can be seen in the supplements. In the revised manuscript, we have added MD95-2037, MD02-2594, IODP1313, GeoB16202-2, DAPC2, BOFS 17K, GeoB9508-1, and the more recent GeoB16206-1 (Voigt et al., 2017) $^{231}\text{Pa}/^{230}\text{Th}$ profiles. The following cores have not been included for the following reasons:

- a) MD03-2705: there is ^{230}Th data but not ^{231}Pa data.
- b) MD02-2588: it is in the Indian Ocean.
- c) GeoB3808-6: the data resolution is too low to be used in this study – there are only five data points over 25–0 ka.

Here the authors made a way better job providing now a thorough overview of the available data. I agree to neglect MD02-2588 and GeoB3808-6 due to the reasons given by the authors. They could still think about including MD03-2705. Its available via

<https://doi.pangaea.de/10.1594/PANGAEA.810309>

because they are right, the ^{231}Pa is curiously not listed in their supplement of Meckler et al.

Response #6: We have included the MD03-2705 $^{231}\text{Pa}/^{230}\text{Th}$ data in the revision.

Reply #10: EW9209-3JPC data points published in Bradtmiller et al. (2014) were based on an old age model (Curry, 1996). As the core's age model was improved with five ^{14}C dates in this study, the $^{231}\text{Pa}/^{230}\text{Th}$ values were re-calculated to correct for radioactive decay of the actinides using the revised sediment ages. The explanation above is included in the revised manuscript.

I welcome this improvement. Please indicate this clearly in the supplement (caption S8) as well.

Response #7: There is a slight mistake in the previous reply #10, and in line 310–311 of the previous manuscript (now revised). EW9209-3JPC age model has nine new ^{14}C dates, not five, and

these new dates are indicated in Supplementary Fig. 10d. We have also indicated the re-calculation of previously published 3JPC data points in Supplementary Fig. 12 and caption.

Reply #11: We agree that overall there is no basin-wide correlation of $^{231}\text{Pa}/^{230}\text{Th}$ with opal in the Atlantic Ocean, as also found in previous time-slice compilation (Bradt Miller et al., 2014; Lippold et al., 2012), as well as in the modern Atlantic (Hayes et al., 2015b). This observation is likely due to the role of AMOC in controlling Atlantic $^{231}\text{Pa}/^{230}\text{Th}$, and potentially the relatively low opal production in the Atlantic. However, ^{231}Pa is more effectively scavenged by opal, an effect shown in experimental studies (Geibert and Usbeck, 2004; Guo et al., 2002; Roberts et al., 2009) and modern observations (Chase et al., 2002; Luo and Ku, 1999; Walter et al., 1997). We feel that we cannot simply dismiss the influence of opal scavenging on past $^{231}\text{Pa}/^{230}\text{Th}$. Including records where there is a significant correlation to opal might add needless complications to the interpretation of past AMOC changes. We have identified $^{231}\text{Pa}/^{230}\text{Th}$ records that are influenced by opal scavenging using the linear correlation method, and the choice of criteria ($r > 0.6$) has been explained in detail in the Supplementary Method. Our approach is efficient and straightforward, although we acknowledge that it has limitations. The reviewer is concerned that important paleoceanographic insight will be lost. However, we have made a new Supplementary Figure 2 which shows that including the opal-influenced $^{231}\text{Pa}/^{230}\text{Th}$ records does not change our main interpretations of AMOC: 1) coherent deglacial changes in the western circulation, 2) two-step AMOC reduction during HS1, 3) some contrasting behaviour between the East and West Atlantic. For our final compilation (Fig. 2), we continue to exclude the opal-influenced $^{231}\text{Pa}/^{230}\text{Th}$ records because: 1) the resulting dataset does not undermine the main paleoceanographic insight presented in this study, 2) we still believe that our approach minimises the potential overprint of (opal) scavenging in sedimentary $^{231}\text{Pa}/^{230}\text{Th}$. The original line 42 is now not appropriate, and has been revised.

I still consider the here applied criterion as somehow arbitrary and selective. As the authors point out with the new Supplementary Figure 2, when including $^{231}\text{Pa}/^{230}\text{Th}$ records which are influenced by opal in their opinion, this not change the main interpretation. This gives rise to the assumption that the criterion is not full developed.

When a linear correlation of 0.6 is a general applicable criterion, I wonder why one locations falls within this selection, while a very adjacent location doesn't, given that Pa/Th records large scale effects. This is the case for e.g.: [25] and [31], [11] and [24], [1] and [14]. However, it is up to them if they unnecessarily aim for reducing the available information.

Response #8: While $^{231}\text{Pa}/^{230}\text{Th}$ could record large-scale oceanic transport signature, it could be modified by opal scavenging at a relatively local scale, given that opal can significantly reduce the residence time of ^{231}Pa in seawater. The cores listed by the reviewer above are situated at or proximal to the continental margins and oceanic plateau: [26] and [32], [12] and [25], [1] and [15] (note the change in core numbers after we included MD03-2705 $^{231}\text{Pa}/^{230}\text{Th}$ data into the compilation – see response #6). Marine primary production (including opal) could have significant variations at margins and over submarine plateau, which might cause local variability in the influence of opal (relative to circulation) on $^{231}\text{Pa}/^{230}\text{Th}$, giving rise to the observed differences in the correlation of $^{231}\text{Pa}/^{230}\text{Th}$ with opal at adjacent sites under such settings. It is not clear that it

will ever be possible to completely unravel competing influences on $^{231}\text{Pa}/^{230}\text{Th}$ (or indeed on any other proxies which have multiple controls) so our approach is to look for and interpret the most robust signals that emerge from multiple cores.

New comments:

e.g. line 125: I cannot see a “gradual increase”. Given the error bars this is a steady peak rather than an increase. Please consider rephrasing.

Response #9: Revised.

e.g. line 131: Here it should be considered and mentioned that the eastern cores are all from a very similar water depth, resulting in very similar $^{231}\text{Pa}/^{230}\text{Th}$ profiles.

Response #10: The eastern cores do not all show similar $^{231}\text{Pa}/^{230}\text{Th}$ profiles (Fig. 2), but we note that eastern cores from similar water depths display the same trend: Core [9] (7° N, 3.43 km) and [13] (3° S, 3.49 km), core [4] (38° N, 3.14 km) and [7] (18° N, 3.09 km), and this observation has been accounted for in the results section.

It should be explained and elaborated what goes on during YD at core GUY14.

Response #11: The $^{231}\text{Pa}/^{230}\text{Th}$ peak of core GUY14 during YD might seem to be more short-lived than the other western cores, but it could well be within the uncertainties associated with age model and bioturbation, especially when there is a reduction in GUY14 sedimentation rate after 12.6 ka (Supplementary Fig. 11). Given these uncertainties, we do not think it is robust to emphasise the $^{231}\text{Pa}/^{230}\text{Th}$ peak of core GUY14 during YD.

Core IODP U1313: Why is this core considered as an eastern core (Fig. 4)?

Response #12: Core IODP U1313 is considered as a Mid-Atlantic Ridge (MAR) core. Figure 4 has been revised accordingly.

Reviewer #2 (Remarks to the Author):

The revised version by Ng et al provides several insights about the composite $^{231}\text{Pa}/^{230}\text{Th}$ observations during the last deglaciation. According to the composite, it is easy to identify phases of deglacial ocean circulation changes. But I'm a bit doubtful whether it can well represent the exact timing when the system went across tipping point if it exists since the signal was smoothed, although I understand that it can provide an overall estimate of past circulation changes. Therefore, caution should be paid to the exact timing of abrupt AMOC shifts.

Response #13: In the revision, we have removed the composite curve from the main text and figures while including it in Supplementary Fig. 6 to address Reviewer 1's concern on the composite's limitations (please see response #3 – #5). Despite not having the composite curve in Fig. 3b, the coherent $^{231}\text{Pa}/^{230}\text{Th}$ observations from the West and deep high-latitude North Atlantic are sufficient to examine AMOC shifts on the millennial timescale, and their relationship with

changes in ice sheet system and climate during the deglacial period. Given the sediment chronology uncertainty, we acknowledge that we could not resolve the timing of AMOC changes on decadal or shorter timescale. Slight revision has been made to line 105–108 accordingly.

Regarding my previous comments, I like the authors' idea to interpret the $^{231}\text{Pa}/^{230}\text{Th}$ records at the eastern sites as the indication of a water-mass transition from the deep Nordic to the Southern Ocean. However, the dynamics regarding the occurrence of YD remains unclear. Other comments are listed in the following:

Response #14: We have improved the discussion on YD (see also response #24). We thank the reviewer for their new comments below, which have helped us with further improving the manuscript.

Abstract: please specify the timing of different steps of deglacial AMOC changes. Otherwise, it is not clear about the timing of the events during the deglaciation for readers.

Response #15: Revised.

Line 24 “reduced” => “weakening”

Response #16: Revised.

Line 37:

add literature Zhang et al (2017 NGS) after “carbon budget”, and Zhang et al (2014 Nature) after “ice sheet volume”, which are relevant and supportive here since they are to explore the AMOC responses to atmospheric CO₂ and ice volume.

Response #17: References added.

Line 54-56:

Here the authors argue that $^{231}\text{Pa}/^{230}\text{Th}$ is proposed to reflect the integrated signal of the overlying ~1km water column when there is active water-mass advection. Then could I consider the $^{231}\text{Pa}/^{230}\text{Th}$ in the deep North Atlantic (lower than ~3.2 km of the water depth) represents the strength of the AABW cell of the AMOC during the LGM, given that the deep Atlantic was bathed by the AABW then and the boundary between GNAIW and AABW ($\delta^{13}\text{C}$ 0.8 isoline is at the water depth of ~ 2.2km in the North Atlantic (Curry and Oppo 2005))? If so, what's the implication for the records in the western sites in the northern North Atlantic?

Response #18: While there was indeed a larger proportion of AABW during the LGM, the deep Atlantic still mainly consisted of mixed AABW and northern-sourced water (Gebbie, 2014). In consideration of the previous finding, we think that the West Atlantic $^{231}\text{Pa}/^{230}\text{Th}$ observations reflect the overall rate of transport of such a mixed deep water mass.

Line 124-125: “a distinct early rise from 18.5 ka to 17 ka, followed by ~2,000 years of a gradual increase to peak values from 17 to 15 ka (Fig. 3b).”

What're the criteria to define the boundary between the two phases of HS1? From my point of view, the boundary should be around 16.2ka BP. Several reasons (not all) are listed in the following: 1) Before that, the composite $^{231}\text{Pa}/^{230}\text{Th}$ increased gradually (Big Dry episode),

followed by an abrupt rise at around 16.2ka BP associated with a shut-down AMOC (Big Wet episode). 2) melting water contribution of the Eurasian ice sheet is trivial after 16.2ka BP, indicating a major role of LIS on the second phase of HS1. 3) Of the $^{231}\text{Pa}/^{230}\text{Th}$ at the Iberian margin the gradual rise ends at around 16.2ka BP. 4) rate of increases in atmospheric CO_2 is evident in the first phase, followed by a CO_2 plateau in the second phase.

Response #19: We thank the reviewer for this nuanced description, and have taken their suggestion. The revised timing of early HS1 and late HS1 are 19–16.5 ka and 16.5–15 ka respectively. We think that 16.5 ka fits all four reasons outlined by the reviewer better than 16.2 ka (Fig. 3, Supplementary Fig. 6).

Line 175-178: 'The observed ... freshwater forcing':

Visually, $^{231}\text{Pa}/^{230}\text{Th}$ slightly decreases from 20ka to 19ka BP, indicating a strengthening AMOC that can lead to a negative mass balance of Eurasian Ice sheet and an increase in the fluvial drainage. Once the freshwater input hits the peak at around 19ka, the AMOC starts to decrease, followed by subsurface warming and IRD release from Eurasian Ice Sheet.

Response #20: This is an interesting point. However, the magnitude of the slight $^{231}\text{Pa}/^{230}\text{Th}$ decrease (0.005) from 20–19 ka observed in the composite record (Supplementary Fig. 6) is well within the analytical uncertainties of the individual data points (up to ± 0.007) during that period. Given this uncertainty, we prefer not to over-interpret the $^{231}\text{Pa}/^{230}\text{Th}$ observation.

Line 178-180: 'Reduced AMOC ... (Fig. 3a)'

The reference is not quite proper since it is specific to ice-shelves from Laurentide Ice Sheet. Additionally, it is not clear to me why the marine-based Eurasian ice shelves are more sensitive to the subsurface warming than the Laurentide ice sheet as the AMOC slows down. Probably, it is associated with the location of the freshwater input (NE North Atlantic), which is more effective on modulating the vertical structure of Nordic sea than the Labrador Sea. The authors need to discuss a bit on this issue.

Response #21: A new coupled atmosphere-ocean-vegetation climate simulation (Ivanovic et al., in review) forced by realistic ice sheet meltwater fluxes has found that a reduction in AMOC linked to accelerated Eurasian ice sheet melting during the last glacial termination from 19–18 ka (Hughes et al., 2016) could lead to sufficient subsurface warming in the Nordic and Arctic Seas to destabilise proximal Eurasian ice sheet margins. Interestingly, the simulated warming does not extend far into the Labrador Sea, which may be the reason that the Laurentide ice sheet, with margins proximal to the Labrador Sea, is less affected by the AMOC slowdown than the Eurasian ice sheets. In the simulation, the main driving factor responsible for the observations above is the magnitude of ice sheet melt, but not the location of meltwater inputs. That aside, the authors think that subsequent climate feedbacks that have not been accounted for in the model (such as melting Eurasian icebergs considered in this study, Fig. 3a) might contribute to further weakening of AMOC, leading to more widespread subsurface warming that could reach the Laurentide ice sheet margins at the Labrador Sea. A more thorough examination of this topic is presented in Ivanovic et al. (in review). For the manuscript presented here, we have added the reference above and made some revisions to the discussion.

Line 188-190: 'This AMOC ... period'

The defined boundary between the two phases does not fit the timing of REF37. Please refer to my comments regarding the definition.

Response #22: Please see response #19.

Line 198-202: 'The timing of AMOC ... ice sheets'

This argument in this sentence is a bit strong. The IRD records just represent the calving part of the ice sheets, and the BIT index only associated with meltwater from the Eurasian ice sheet. All of them cannot rule out the existence of persistent meltwater from Laurentide ice sheet during BA. Furthermore, within the age uncertainty, there is a rapid sea level rise at the same time, which indicates that BA warming is accompanied with freshwater input, in contrast to the argument here. One alternative explanation for BA warming associated with AMOC resumption is due to the increased atmospheric CO₂ (Zhang et al 2017 NGS), since the increasing CO₂ can overcome the effect of freshwater input on the AMOC strength.

Response #23: We agree with the reviewer's points and have made revision to the discussion on BA mechanism.

Line 211-214: 'instead ... sea-ice model'

"most likely" is a bit strong here. As I pointed out in last time, it is not that simple to provide a reasonable triggering mechanism for YD (please see a potential new triggering mechanism of YD published in CPD recently -- Baldini et al 2017 CPD). The IRD record from New Found land indeed indicates a contemporary melting iceberg that sustains the cold condition. However, the amount of meltwater input from Icebergs could be smaller and shorter than that during HS1 (e.g. ~0.09Sv for 300 years during YD [Tarasov and Peltier 2005]), which is hard to explain the 1200-year-long YD event by coupled GCMs. Evidence regarding the timing of IRD events also indicate that iceberg discharge occurred after the start of the YD (Hillaire-Marcel and Bilodeau, 2000). The concept of AMOC stability can potentially overcome this problem. I'm a bit disappointed as the AMOC stability is not mentioned in this context. From my point of view, YD is a good testbed to evaluate its linkage to AMOC

stability associated with ice volume changes (Zhang et al 2014), as the global mean sea level is just at its intermediate level during the glacial-interglacial cycles. This indicates that the climate system under the YD background condition is more sensitive to forcing that can modulate the strength of the AMOC. Therefore, even a temporal forcing under this context can lead to a long-term persistent consequence on the climate system.

Response #24: We acknowledge that the triggering mechanism for YD remains an open question. We recognise the significance to account for the AMOC stability under climate conditions such as those during the YD, and have made revision to the discussion on YD accordingly.

Line 235:

'ref. 51': is not proper here. In the abstract of REF 51, it proposed that the interface shoals but the vertical volume of NADW are not different from modern day, which is a contrast to "weakened" as authors stated here.

Response #25: We have removed this sentence in the manuscript.

Line 236:

'ref. 47' is not proper here. (This paper is to discuss interglacial rather glacial climate)

Response #26: We have removed this sentence in the manuscript.

Line 249-250:

Nice point here. So, is it plausible that ocean circulation in the upper level (above 3km) can be stronger during the LGM than the modern day in the North Atlantic? what's its implication for the western cores?

Response #27: The $^{231}\text{Pa}/^{230}\text{Th}$ data suggest a more rapid water mass transport for the eastern sites above 3 km water depth during the LGM than the modern day, and the opposite for the western sites. Establishing the extent of this east-west contrast in circulation history will require further data and model testing in future ocean circulation modelling studies.

Reviewer #3 (Remarks to the Author):

2nd review of Ng et al.

The authors have taken into account some of the comments that were made during the 1st review. I think the manuscript has improved and reads better than in the first version. Overall, I think it is a welcome addition and a nice paper, for which I recommend publication after minor revisions. However, I should point out that I am still unconvinced by some of the interpretation made, particularly with respect to the East-West contrast.

Response #28: We thank the reviewer for acknowledging the improvements made in the first revision, and their recommendation for publication after minor revisions. Below we have addressed their outstanding concerns.

1) The authors suggest that deep-water circulation was stronger in the Northeast Atlantic than in the Northwest. They further suggest this could be due to changes in the location of deep water formation. However, even if deep water was only formed in the Nordic Sea, it would still flow into the western part of the Atlantic as a deep western boundary current and a minor part of the flow would go into the Eastern part of the basin. As detailed below, there is little evidence to suggest that the eastern equatorial Atlantic cores record northern sourced deep water masses. Therefore the East-West discrepancy mostly relies on the Iberian margin core (and a few points from MAR).

Response #29: We have included an additional $^{231}\text{Pa}/^{230}\text{Th}$ record from off Mauritania in the eastern basin (see response #6). The core off Mauritania (18° N, 3.09 km) displays a very similar trend as the Iberian margin core (38° N, 3.14 km) (Fig. 2b), providing further support to the East-West difference in deep water transport during the LGM. In addition, there are other supporting evidence for a Nordic deep water in the eastern basin during the LGM (Meland et al., 2008; Yu et al., 2008), but no such evidence is found in the western basin. We are also intrigued by the flow

path of this potential Nordic deep water – maybe it entered the North Atlantic through the Iceland-Scotland Ridge into the eastern basin instead of the shallower Denmark Strait into the western basin if the deep water was relatively dense due to the increased brine rejection in the Nordic Seas during the LGM (Meland et al., 2008). Further testing of this hypothesis will be required in future ocean circulation modelling studies.

2) I am not convinced about interpretation of cores 8 and 12 (GVY01 and RC24). The relationship between Pa/Th has been tested in the Northwest Atlantic (Rempfer et al., 2017), but I am not sure of the AMOC influence on the eastern equatorial Atlantic. Assuming there is a deep Northeast water mass flowing south, then it should be weaker at GVV01 than at the Iberian Margin. But the relationship between changes in these cores and the Iberian margin is weak. In addition, this core could also be under the influence of Antarctic Bottom Water (AABW), but AABW would first influence the Brazilian margin (cores 9 and 10), before these ones. Therefore, contrarily to what expected these 2 cores show the lowest Pa/Th values across the deglaciation and the smallest changes. This means that these 2 cores most likely reflect something else than North Atlantic deep water masses or AABW.

Response #30: Low $^{231}\text{Pa}/^{230}\text{Th}$ values well below the production ratio observed in GVV01 (7° N, 3.43 km) and RC24-12 (3° S, 3.49 km) (Fig. 2c) require a mechanism that maintains a net export of ^{231}Pa from these sites over the last 25 kyr. Possible mechanism for these sites include eddy diffusion associated with boundary scavenging (Hayes et al., 2015), horizontal transport driven by coastal upwelling, or/and deep water mass transport. For the former two processes, we would expect a net accumulation of ^{231}Pa giving rise to $^{231}\text{Pa}/^{230}\text{Th}$ values well above the production ratio at the African margin throughout the last 25 kyr including the LGM and HS1. However, such an observation is not made in the African margin record (GeoB9508-5, 15° N, 2.38 km) (Supplementary Fig. 2) proximal to GVV01 and RC24-12. Therefore, boundary scavenging and coastal upwelling are not sufficient to explain the $^{231}\text{Pa}/^{230}\text{Th}$ observations at these sites. Whilst we agree that low-latitude East Atlantic sites are less sensitive to AMOC compared to the North-West Atlantic sites (Rempfer et al., 2017), a net export of ^{231}Pa from GVV01 and RC24-12 are driven by NADW during the Holocene (Bradtmiller et al., 2014), and could be sustained by the proposed Nordic deep water (please see response #29) during the LGM. We also agree that some questions are left to be answered regarding the identity of deep water mass that could maintain the ^{231}Pa export at these two sites during HS1. These questions warrant future investigation with more $^{231}\text{Pa}/^{230}\text{Th}$ reconstructions and modelling studies.

3) Two phases of HS1: the authors distinguish 2 phases for HS1, an early and late HS1. If the authors really want to define phases for HS1, then maybe a bit more thought and/or justification should be put into that effort.

i) The authors never actually give time-frames for these 2 periods (e.g. from 18.6 to 17ka and from 17 to 15.4ka), which are only seen in Figure 3.

Response #31: We have added this information in the revision (for example, line 176–178). The revised timing of early HS1 and late HS1 are 19–16.5 ka and 16.5–15 ka respectively (please see response #19).

ii) These two phases are separated by the increase in BIT index (at 17ka) and more or less coincide with the Pa/Th record. The early phase broadly refers to the time period where Pa/Th increases and where some IRD peaks are seen in the Rockall Basin and fluvial discharges in the Bay of Biscay. But the composite Pa/Th reaches almost maximal values at 17.5 ka, meaning that the AMOC is most likely very weak starting from that point on, with important implications for the climate and carbon cycle. Shouldn't the early phase stop at 17.5ka?

Response #32: Reviewer 2 has put forward several points in defining the boundary between the two phases of HS1. We agree with their points and have re-defined the boundary at 16.5 ka (please see response #19).

iii) There is another slight increase in Pa/Th at 16ka. The authors mention this could be coincident with observations of changes in the hydrological cycle. But I wonder, are these changes in Pa/Th really significant? Do the authors really think that this change at 16ka really represent a change in the formation and transport of deep-water mass?

Response #33: We think that the $^{231}\text{Pa}/^{230}\text{Th}$ observations at ~ 16.5 ka are potentially linked with changes in the hydrological cycle at around the same period (Broecker and Putnam, 2012). Although the western sites have reached peak values of $^{231}\text{Pa}/^{230}\text{Th}$ (Fig. 2a & 3b), the deep North-East Atlantic sites show drastic increase in $^{231}\text{Pa}/^{230}\text{Th}$ at this time (Fig. 2b), signifying a further weakening of deep circulation and deep water formation that represent the second phase of AMOC reduction.

iv) The recovery from HS1 (15.5-14.5 ka) is not labeled as anything.... Maybe late HS1 should be extended to ~ 14.7 ka, when NGRIP d18O increases abruptly.

Response #34: The grey shading (Fig. 3) that mark the HS1-BA transition (recovery) and YD-early Holocene transition (recovery) are defined in the figure caption. In the revision we have extended late HS1 to 15 ka.

Author's references

Bradt Miller, L.I., McManus, J.F., Robinson, L.F., 2014. $^{231}\text{Pa}/^{230}\text{Th}$ evidence for a weakened but persistent Atlantic meridional overturning circulation during Heinrich Stadial 1. *Nat Commun* 5.

Broecker, W., Putnam, A.E., 2012. How did the hydrologic cycle respond to the two-phase mystery interval? *Quaternary Sci Rev* 57, 17-25.

Gebbie, G., 2014. How much did Glacial North Atlantic Water shoal? *Paleoceanography* 29, 190-209.

Hayes, C.T., Anderson, R.F., Fleisher, M.Q., Huang, K.F., Robinson, L.F., Lu, Y.B., Cheng, H., Edwards, R.L., Moran, S.B., 2015. Th-230 and Pa-231 on GEOTRACES GA03, the US GEOTRACES North Atlantic transect, and implications for modern and paleoceanographic chemical fluxes. *Deep-Sea Res Pt II* 116, 29-41.

Hughes, A.L.C., Gyllencreutz, R., Lohne, Ø.S., Mangerud, J., Svendsen, J.I., 2016. The last Eurasian ice sheets - a chronological database and time-slice reconstruction, DATED-1. *Boreas* 45, 1-45.

Ivanovic, R.F., Gregoire, L.J., Burke, A., Wickert, A.D., Valdes, P.J., Ng, H.C., Robinson, L.F., McManus, J.F., Mitrovica, J.X., Lee, L., Dentith, J.E., in review. Acceleration of northern ice sheet melt induces AMOC slowdown and northern cooling in simulations of the early last deglaciation. *Paleoceanography and Paleoclimatology*.

Lippold, J., Gherardi, J.M., Luo, Y.M., 2011. Testing the Pa-231/Th-230 paleocirculation proxy: A data versus 2D model comparison. *Geophys Res Lett* 38, L20603.

Meland, M.Y., Dokken, T.M., Jansen, E., Hevroy, K., 2008. Water mass properties and exchange between the Nordic seas and the northern North Atlantic during the period 23-6 ka: Benthic oxygen isotopic evidence. *Paleoceanography* 23, PA1210.

Rempfer, J., Stocker, T.F., Joos, F., Lippold, J., Jaccard, S.L., 2017. New insights into cycling of ^{231}Pa and ^{230}Th in the Atlantic Ocean. *Earth Planet Sc Lett* 468, 27-37.

Yu, J.M., Elderfield, H., Piotrowski, A.M., 2008. Seawater carbonate ion-delta C-13 systematics and application to glacial-interglacial North Atlantic ocean circulation. *Earth Planet Sc Lett* 271, 209-220.

Reviewers' comments:

Reviewer #1 (Remarks to the Author):

This is my third review of this manuscript. The authors have improved the manuscript and settled most of my concerns. There are still a few issues I would appreciate if have been approached differently and results I would have interpreted in a different way, but I'm aware that these points may be subject to a personal opinion and do not strictly contradict the state-of-the-art. Therefore, I recommend publication after considering a few final points.

I'm still not happy with the "composite 231Pa/230Th" being part of this paper. I'm afraid that we might never come to a perfect agreement on this point, but I appreciate that the authors found kind of a compromise by moving it to the supplement, and more important, by adding some explaining comments.

One of two points, which still require improvement is the criterion of rejecting 231Pa/230Th records due to correlations with opal fluxes. I'm not satisfied by the author's response #8 (see below).

Authors' response #8: "While 231Pa/230Th could record large-scale oceanic transport signature, it could be modified by opal scavenging at a relatively local scale, given that opal can significantly reduce the residence time of 231Pa in seawater. The cores listed by the reviewer above are situated at or proximal to the continental margins and oceanic plateau: [26] and [32], [12] and [25], [1] and [15] (note the change in core numbers after we included MD03-2705 231Pa/230Th data into the compilation – see response #6). Marine primary production (including opal) could have significant variations at margins and over submarine plateau, which might cause local variability in the influence of opal (relative to circulation) on 231Pa/230Th, giving rise to the observed differences in the correlation of 231Pa/230Th with opal at adjacent sites under such settings. It is not clear that it will ever be possible to completely unravel competing influences on 231Pa/230Th (or indeed on any other proxies which have multiple controls) so our approach is to look for and interpret the most robust signals that emerge from multiple cores."

I agree with the explanation given in response#8 on the difference in the correlation of opal to 231Pa/230Th in adjacent records. This could now be mentioned in the text. But I still do not agree with the applied criterion. I also do not agree that this criterion ($r > 0.6$) is sufficiently explained in the supplement. First, from a statistical point of view $r > 0.6$ is an arbitrary threshold especially when it comes to describing this correlation as "significant". The authors included p-values in the supplement figures 3-5, but in my copy I was not able to find the consideration (as well as no mentioning in the captions) of the p-values. Before "stigmatising" a 231Pa/230Th record as opal-driven in Supplement-table-1 the p-value should be considered as well. While the $r > 0.6$ indicates a correlation, this correlation is not necessarily significant. Thus, please remove "significant" from the column header in in Supplement-table-1.

We all agree on that opal has an influence of 231Pa/230Th. But the recorded and measured opal flux may provide little information on the actual opal flux once scavenged the 231Pa, simply due to significant opal dissolution (e.g. Demaster2002). So what is correlated here is the preserved opal with 231Pa/230Th. But during the time of largest variations in AMOC and thus 231Pa/230Th the water mass changed from a well opal preserving (SSW) towards an opal-corrosive (NSW) water mass. So, the observed correlation is predominantly just a correlation and not a chain of causality. Thus, if the 0.6 criterion is still kept in the manuscript this point should be mentioned, as well as that the criterion is more or less selected to be on the very very safe side and words like "significant" should be avoided. I think this is an important point in particular because (as the authors state) "...including the opal-influenced 231Pa/230Th records does not change our main interpretations of AMOC: 1) coherent deglacial changes in the western circulation, 2) two-step AMOC reduction during HS1, 3) some contrasting behaviour between the East and West Atlantic. For our final compilation (Fig. 2),", which may indicate that the $r > 0.6$ criterion is too exclusionary.

Former comment: It should be explained and elaborated what goes on during YD at core GVV14.

Authors' response #11: "The $^{231}\text{Pa}/^{230}\text{Th}$ peak of core GVV14 during YD might seem to be more short-lived than the other western cores, but it could well be within the uncertainties associated with age model and bioturbation, especially when there is a reduction in GVV14 sedimentation rate after 12.6 ka (Supplementary Fig. 11). Given these uncertainties, we do not think it is robust to emphasise the $^{231}\text{Pa}/^{230}\text{Th}$ peak of core GVV14 during YD."

I'm not sure if I misunderstand the response or if I should have explained my point better, because this point is not solved in my opinion. GVV14 shows a general low sedimentation rates, but until 13.3 kyrs sed-rate is extremely low. Then a sudden jump by a factor of 37(!) occurs lasting for only 0.7 ka, when sed-rates return to ~ 1.4 cm/yrs. I want to encourage the authors to go here into the very details of each measurements and calibrations involved here. But independent from that $^{231}\text{Pa}/^{230}\text{Th}$ also behaves strange with two extreme values not seen in this record since the LGM. Again I want to encourage the authors to re-check the raw data carefully here. Else such sudden peaks would question the analytical significance of the error bars applied here. On the other side, if these measurements are robust, what would this mean? There is an extreme short-lived rise in the middle of the YD in $^{231}\text{Pa}/^{230}\text{Th}$? Does this have any paleoceanographic implications? We would expect higher $^{231}\text{Pa}/^{230}\text{Th}$ during YD (McManus2004), but so steep and short? Bioturbation may not account for this, because the sed-rate is especially high here, minimising the effects of bioturbation. Or is this sharp signal preserved due to the high (if true) sed-rate and faded out by the low sed-rate after and before this event? Would this be possible? Another related (graphical) issue is the presentation in Fig.2. It somehow looks like that there are three high $^{231}\text{Pa}/^{230}\text{Th}$ values at GVV14 during YD, which is just an artefact due to the broad lines and symbols rims. Maybe better put the lines in a layer below the individual symbols and decrease their width.

Reviewer #2 (Remarks to the Author):

The authors provide clear answers to my previous comments and I recommend to publish in Nature Comm after a minor revision as mentioned below:

Line 233: "Under YD climate conditions" => Under an intermediate ice volume like YD.

Again, this is nice work that provides in-depth understanding of AMOC changes during early deglaciation. Congratulations to the authors.

Reviewer #3 (Remarks to the Author):

3rd Review of Ng et al.,

Ng et al. present an interesting and important study on deglacial changes in oceanic circulation based on Pa/Th.

Please find a few minor points below.

o I am not at ease with the definition/timing of the two phases of HS1.

It is crucial to obtain a better grasp on the deglacial AMOC evolution. Indeed, this would allow a better understanding of the climatic and global carbon cycle changes occurring during the

deglaciation. This is in this context that this new study is important and interesting.

However, the rationale behind the definition is still not clear. Reviewer 2 suggested to split HS1 at 16.2 ka based on climatic and carbon cycle changes occurring during that time. This is fine but I am not completely convinced that this is really what is coming out of the data. In this version the authors settle for a "middle ground" by splitting it at 16.5 ka.

While I agree that there is coherence in the data, the data is also noisy in the details. The authors had found a way around it by making a composite, which is now in SI because there are obvious issues associated with making such a composite as pointed out by Reviewer 1.

However, if we look at this composite, there is a Pa/Th increase between 19 and 17.5ka, followed by a plateau at almost maximum value and arguably another small increase at ~16ka.

When looking at individual records, the maximum value is obtained somewhere between 18 and 16.2ka: the Bermuda rise core [5] and Rockall basin core [1], which are in the direct path of (both a potential Western and Eastern) NADW reach maximum value at 17.5 ka, Core nb [12] on the Brazil margin reaches max values at 17ka, and indeed the Iberian margin core only reaches max values at ~16.2ka, but note that in core [7] only 1 measurement at 15.5 ka shows a decrease: all this simply implies that only relying on one core can be misleading and would lead to different timing of the phases of HS1.

The two proposed HS1 phases somehow fit with the IRD plots shown, even though the IRD does start to increase at 17ka in B. It should however be noted that for example core CH69-K09 (Cortijo et al., 1999), which is located near core [2] (~42N, 47W), and thus under Laurentide ice-sheet influence, displays an IRD peak between 17.9 and 16.7 ka.

My query is thus for the authors to objectively look at their data and decide what can be inferred from their data within the dating/measurements errors and other local effects. 2 phases could be changed into 3 with a discussion of errors and limitations: 1) 19 to 17.5 ka: gradual AMOC decrease associated with Eurasian ice-sheet melting, 2) 17.5-16.5 (or 16.2ka): very weak AMOC with both Eurasian and Laurentide ice-sheet disintegrations, 3) 16.5 to 15ka: AMOC very weak, potentially even weaker than in phase 2, associated with Laurentide ice-sheet melting.

o L. 187-188: IRD from Laurentide ice sheet should be 17 to 15ka in this context.

o This is in contrast with the other part of this paragraph (L. 170-179), where "peaks" of IRD are defined, that are in broad agreement with the different phases.

o L. 204: I am a bit surprised that the transient modeling simulations of the last deglaciation (e.g. Liu et al., 2009, Menviel et al., 2011), which highlight the role of AMOC in millennial -scale variability are never mentioned. It would have been appropriate to cite them here.

o L. 209-211 might need rephrasing.

o L. 223-225: Modelling studies have also highlighted the role of AMOC weakening during the YD (e.g. Menviel et al., 2011, Renssen et al., 2015).

" L. 259-261: I don't think this sentence is correct. On p10 of Gebbie 2014, it is mentioned that the modeling results suggest an unchanged LGM volumetric census in both the western and eastern Atlantic.

Also if the origin of the deep eastern Atlantic water is the Nordic Sea, then why is it not visible in site [1]?

o Figure 3b: I would suggest to remove cores [8] and [10] as they don't add much information but contribute in making the figure less clear.

Response to reviewers

Key:

- Reviewers' comments
- Author's response

Reviewer #1 (Remarks to the Author):

This is my third review of this manuscript. The authors have improved the manuscript and settled most of my concerns. There are still a few issues I would appreciate if have been approached differently and results I would have interpreted in a different way, but I'm aware that these points may be subject to a personal opinion and do not strictly contradict the state-of-the-art. Therefore, I recommend publication after considering a few final points.

Response #1: We are very pleased that the reviewer is supportive of publication of the paper, recognising that we may have some slight differences in our final interpretation. We have taken the additional points and clarifications on board and modified the text accordingly as described below.

I'm still not happy with the "composite $^{231}\text{Pa}/^{230}\text{Th}$ " being part of this paper. I'm afraid that we might never come to a perfect agreement on this point, but I appreciate that the authors found kind of a compromise by moving it to the supplement, and more important, by adding some explaining comments.

Response #2: We followed the editor's suggestion on the approach of including the composite and explanatory comments in the supplementary material. We appreciate that the reviewer has a different opinion on the value of composite $^{231}\text{Pa}/^{230}\text{Th}$ record, however they do acknowledge that a compromise has been reached on this point which is acceptable to all parties.

One of two points, which still require improvement is the criterion of rejecting $^{231}\text{Pa}/^{230}\text{Th}$ records due to correlations with opal fluxes. I'm not satisfied by the author's response #8 (see below).

Authors' response #8: "While $^{231}\text{Pa}/^{230}\text{Th}$ could record large-scale oceanic transport signature, it could be modified by opal scavenging at a relatively local scale, given that opal can significantly reduce the residence time of ^{231}Pa in seawater. The cores listed by the reviewer above are situated at or proximal to the continental margins and oceanic plateau: [26] and [32], [12] and [25], [1] and [15] (note the change in core numbers after we included MD03-2705 $^{231}\text{Pa}/^{230}\text{Th}$ data into the compilation – see response #6). Marine primary production (including opal) could have significant variations at margins and over submarine plateau, which might cause local variability in the influence of opal (relative to circulation) on $^{231}\text{Pa}/^{230}\text{Th}$, giving rise to the observed differences in the correlation of $^{231}\text{Pa}/^{230}\text{Th}$ with opal at adjacent sites under such settings. It is not clear that it will ever be possible to completely unravel competing influences on $^{231}\text{Pa}/^{230}\text{Th}$

(or indeed on any other proxies which have multiple controls) so our approach is to look for and interpret the most robust signals that emerge from multiple cores.”

I agree with the explanation given in response#8 on the difference in the correlation of opal to $^{231}\text{Pa}/^{230}\text{Th}$ in adjacent records. This could now be mentioned in the text.

Response #3: We have mentioned this point in the text.

But I still do not agree with the applied criterion. I also do not agree that this criterion ($r > 0.6$) is sufficiently explained in the supplement. First, from a statistical point of view $r > 0.6$ is an arbitrary threshold especially when it comes to describing this correlation as “significant”. The authors included p-values in the supplement figures 3-5, but in my copy I was not able to find the consideration (as well as no mentioning in the captions) of the p-values. Before “stigmatising” a $^{231}\text{Pa}/^{230}\text{Th}$ record as opal-driven in Supplement-table-1 the p-value should be considered as well. While the $r > 0.6$ indicates a correlation, this correlation is not necessarily significant. Thus, please remove “significant” from the column header in Supplement-table-1.

Response #4: As the reviewer notes below, there is agreement in the literature that opal influences $^{231}\text{Pa}/^{230}\text{Th}$ ratios in the ocean (water, particles and sediments), given the role of opal in the scavenging of the radionuclides. The GEOTRACES program is adding greatly to this knowledge, and we have included this new knowledge in our paper. However, there is still no consensus on an objective cut-off criterion to define the impact of opal content on $^{231}\text{Pa}/^{230}\text{Th}$ – in fact there may never be, given the complex processes at work in the oceans. Despite the controls of opal and other scavenging processes, we still see coherent behaviours of $^{231}\text{Pa}/^{230}\text{Th}$ in the Atlantic sediment cores, which can be used to infer past ocean circulation. We chose a conservative r value of 0.6 as a transparent, easily traceable criterion that best fits the current knowledge. We have included the p-value information in Supplementary Table 1 and included consideration of p-values in the Supplementary Method to assess the statistical significance of the correlations. The word “significant” has been removed from the column header in Supplementary Table 1. We have added further clarification to the choice of our approach and the applied criterion ($r > 0.6$) in the Supplementary Method.

We all agree on that opal has an influence of $^{231}\text{Pa}/^{230}\text{Th}$. But the recorded and measured opal flux may provide little information on the actual opal flux once scavenged the ^{231}Pa , simply due to significant opal dissolution (e.g. Demaster2002). So what is correlated here is the preserved opal with $^{231}\text{Pa}/^{230}\text{Th}$. But during the time of largest variations in AMOC and thus $^{231}\text{Pa}/^{230}\text{Th}$ the water mass changed from a well opal preserving (SSW) towards an opal-corrosive (NSW) water mass. So, the observed correlation is predominantly just a correlation and not a chain of causality. Thus, if the 0.6 criterion is still kept in the manuscript this point should be mentioned, as well as that the criterion is more or less selected to be on the very very safe side and words like “significant” should be avoided. I think this is an important point in particular because (as the authors state) “...including the opal-influenced $^{231}\text{Pa}/^{230}\text{Th}$ records does not change our main interpretations of AMOC:

1) coherent deglacial changes in the western circulation, 2) two-step AMOC reduction during HS1, 3) some contrasting behaviour between the East and West Atlantic. For our final compilation (Fig. 2),”, which may indicate that the $r > 0.6$ criterion is too exclusionary.

Response #5: Again, we concur with the reviewer that there is widespread agreement on the influence opal has on $^{231}\text{Pa}/^{230}\text{Th}$, and it has been long known that there is post-depositional opal

dissolution. The measured opal concentration in the sediment is indeed the preserved opal content, and so does not fully reflect the sinking particle flux. Our choice of having r value of 0.6 as a cut-off criterion may indeed lead to inclusion of cores where opal has been lost, and cores where opal was prevalent but circulation still dominated the $^{231}\text{Pa}/^{230}\text{Th}$ signal. It is impossible to guard against such possibilities in the natural setting. However, we have selected a transparent criterion which is easy to use and based on observational evidence. We have clarified our position in the paper, adding explicit discussion of the issue of post-depositional opal dissolution. We have acknowledged that the approach of excluding all cores that show $r > 0.6$ for the opal flux– $^{231}\text{Pa}/^{230}\text{Th}$ correlation from further interpretation of Atlantic circulation changes is indeed a conservative one, but it is the most efficient and straightforward approach to minimise the potential overprint of opal scavenging in sedimentary $^{231}\text{Pa}/^{230}\text{Th}$ observations.

Former comment: It should be explained and elaborated what goes on during YD at core GVV14.

Authors' response #11: "The $^{231}\text{Pa}/^{230}\text{Th}$ peak of core GVV14 during YD might seem to be more short-lived than the other western cores, but it could well be within the uncertainties associated with age model and bioturbation, especially when there is a reduction in GVV14 sedimentation rate after 12.6 ka (Supplementary Fig. 11). Given these uncertainties, we do not think it is robust to emphasise the $^{231}\text{Pa}/^{230}\text{Th}$ peak of core GVV14 during YD."

I'm not sure if I misunderstand the response or if I should have explained my point better, because this point is not solved in my opinion. GVV14 shows a general low sedimentation rates, but until 13.3 kyrs sed-rate is extremely low. Then a sudden jump by a factor of 37(!) occurs lasting for only 0.7 ka, when sed-rates return to ~ 1.4 cm/yr. I want to encourage the authors to go here into the very details of each measurements and calibrations involved here. But independent from that $^{231}\text{Pa}/^{230}\text{Th}$ also behaves strange with two extreme values not seen in this record since the LGM. Again I want to encourage the authors to re-check the raw data carefully here. Else such sudden peaks would question the analytical significance of the error bars applied here.

Response #6: We can confirm that the $^{231}\text{Pa}/^{230}\text{Th}$ measurements, the ^{14}C age measurements and calibrations of GVV14 are correct. In addition, based on GVV14 core scan and XRF data, no abrupt shifts in sediment properties and elemental composition are observed over the YD interval, indicating that the short-lived $^{231}\text{Pa}/^{230}\text{Th}$ peak is not an outlier related to sudden change in sediment composition.

On the other side, if these measurements are robust, what would this mean? There is an extreme short-lived rise in the middle of the YD in $^{231}\text{Pa}/^{230}\text{Th}$? Does this have any paleoceanographic implications? We would expect higher $^{231}\text{Pa}/^{230}\text{Th}$ during YD (McManus2004), but so steep and short? Bioturbation may not account for this, because the sed-rate is especially high here, minimising the effects of bioturbation. Or is this sharp signal preserved due to the high (if true) sed-rate and faded out by the low sed-rate after and before this event? Would this be possible?

Response #7: We have included discussion on the implications of the short-lived $^{231}\text{Pa}/^{230}\text{Th}$ peak observed in core GVV14 at the YD interval. The $^{231}\text{Pa}/^{230}\text{Th}$ peak suggests that there could have been an abrupt change in water mass transport at this site during the cold stadial. The YD peak of $^{231}\text{Pa}/^{230}\text{Th}$ observed here appears to be short-lived relative to the other cores (Fig. 2), potentially implying some spatial differences in the timing and duration of the reduction in water mass transport during the YD decline of AMOC. However, we cannot rule out that combined effects of

bioturbation and drop in sedimentation rate after 12.6 ka (Supplementary Fig. 11) in the GVV14 core might have mixed away the latter part of the $^{231}\text{Pa}/^{230}\text{Th}$ peak, which would underestimate the duration of the signal. Further constraining the Atlantic $^{231}\text{Pa}/^{230}\text{Th}$ trends during YD will require additional cores with high sedimentation rate throughout the cold stadial.

Another related (graphical) issue is the presentation in Fig.2. It somehow looks like that there are three high $^{231}\text{Pa}/^{230}\text{Th}$ values at GVV14 during YD, which is just an artefact due to the broad lines and symbols rims. Maybe better put the lines in a layer below the individual symbols and decrease their width.

Response #8: We have improved the clarity of the figure.

Reviewer #2 (Remarks to the Author):

The authors provide clear answers to my previous comments and I recommend to publish in Nature Comm after a minor revision as mentioned below:

Line 233: "Under YD climate conditions" => Under an intermediate ice volume like YD.

Response #9: Revised.

Again, this is nice work that provides in-depth understanding of AMOC changes during early deglaciation. Congratulations to the authors.

Response #10: We sincerely appreciate all the constructive comments from the reviewer, which have certainly helped to improve the manuscript for publication in Nature Communications. Thank you.

Reviewer #3 (Remarks to the Author):

3rd Review of Ng et al.,

Ng et al. present an interesting and important study on deglacial changes in oceanic circulation based on Pa/Th.

Please find a few minor points below.

Response #11: We thank the reviewer as they continue to find our study interesting and important. Below we have addressed their few remaining minor points.

o I am not at ease with the definition/timing of the two phases of HS1.

It is crucial to obtain a better grasp on the deglacial AMOC evolution. Indeed, this would allow a better understanding of the climatic and global carbon cycle changes occurring during the deglaciation. This is in this context that this new study is important and interesting.

However, the rationale behind the definition is still not clear. Reviewer 2 suggested to split HS1 at 16.2 ka based on climatic and carbon cycle changes occurring during that time. This is fine but I am not completely convinced that this is really what is coming out of the data. In this version the authors settle for a "middle ground" by splitting it at 16.5 ka.

Response #12: We agree with the reviewer that our new study provides further insights into the deglacial AMOC evolution, especially on the HS1 phases, which have important climate implications. With the data available we are able to define a two-phase change in AMOC (Fig. 3b, Supplementary Fig. 6b), meltwater contribution from the Eurasian and Laurentide ice sheets (Fig. 3a), global hydrological cycle (Broecker and Putnam, 2012), and the carbon cycle (Fig. 3e) during HS1. Based on the combined dataset, we think that the boundary between the two phases is best defined at 16.5 ka (Fig. 3, Supplementary Fig. 6).

While I agree that there is coherence in the data, the data is also noisy in the details. The authors had found a way around it by making a composite, which is now in SI because there are obvious issues associated with making such a composite as pointed out by Reviewer 1.

However, if we look at this composite, there is a Pa/Th increase between 19 and 17.5ka, followed by a plateau at almost maximum value and arguably another small increase at ~16ka.

When looking at individual records, the maximum value is obtained somewhere between 18 and 16.2ka: the Bermuda rise core [5] and Rockall basin core [1], which are in the direct path of (both a potential Western and Eastern) NADW reach maximum value at 17.5 ka, Core nb [12] on the Brazil margin reaches max values at 17ka, and indeed the Iberian margin core only reaches max values at ~16.2ka, but note that in core [7] only 1 measurement at 15.5 ka shows a decrease: all this simply implies that only relying on one core can be misleading and would lead to different timing of the phases of HS1.

Response #13: We agree that the timing of the HS1 phases can be biased if we solely rely on $^{231}\text{Pa}/^{230}\text{Th}$ observations of any one core. Instead, we established that all the cores which display pronounced increase in $^{231}\text{Pa}/^{230}\text{Th}$ beginning ~19 ka (cores [1], [5], [11] and [12]) reach maximum values "latest" by ~16.5 ka (Fig. 3b, Supplementary Fig. 6b). In other words, 16.5 ka marks the very end of the first phase of AMOC decline. This is generally consistent with the timing of changes in meltwater contribution from the Eurasian and Laurentide ice sheets (Fig. 3a), global hydrological cycle (Broecker and Putnam, 2012), and the carbon cycle (Fig. 3e), lending support to our decision of defining 16.5 ka as the boundary of two phases in HS1.

The two proposed HS1 phases somehow fit with the IRD plots shown, even though the IRD does start to increase at 17ka in B. It should however be noted that for example core CH69-K09 (Cortijo et al., 1999), which is located near core [2] (~42N, 47W), and thus under Laurentide ice-sheet influence, displays an IRD peak between 17.9 and 16.7 ka.

My query is thus for the authors to objectively look at their data and decide what can be inferred from their data within the dating/measurements errors and other local effects. 2 phases could be changed into 3 with a discussion of errors and limitations: 1) 19 to 17.5 ka: gradual AMOC decrease associated with Eurasian ice-sheet melting, 2) 17.5-16.5 (or 16.2ka): very weak AMOC with both Eurasian and Laurentide ice-sheet disintegrations, 3) 16.5 to 15ka: AMOC very weak, potentially even weaker than in phase 2, associated with Laurentide ice-sheet melting.

Response #14: It would be very interesting if there were indeed three phases, and we agree that the data leaves this possibility open. However, given our current knowledge of chronological uncertainties (Supplementary Fig. 14), it is challenging to robustly establish three phases in HS1.

Meanwhile, we can confidently resolve two phases of AMOC and ice sheet changes during HS1 based on the strongest shared signal: first phase of AMOC decline associated with Eurasian ice sheet melt and ice rafting, and second phase of maximum AMOC reduction coincident with Laurentide ice rafting. Further resolving three HS1 phases of AMOC and climate changes is an interesting hypothesis for future high-resolution studies.

o L. 187-188: IRD from Laurentide ice sheet should be 17 to 15ka in this context.

o This is in contrast with the other part of this paragraph (L. 170-179), where "peaks" of IRD are defined, that are in broad agreement with the different phases.

Response #15: Revised.

o L. 204: I am a bit surprised that the transient modeling simulations of the last deglaciation (e.g. Liu et al., 2009, Menviel et al., 2011), which highlight the role of AMOC in millennial -scale variability are never mentioned. It would have been appropriate to cite them here.

Response #16: References added.

o L. 209-211 might need rephrasing.

Response #17: Rephrased.

o L. 223-225: Modelling studies have also highlighted the role of AMOC weakening during the YD (e.g. Menviel et al., 2011, Renssen et al., 2015).

Response #18: References added.

" L. 259-261: I don't think this sentence is correct. On p10 of Gebbie 2014, it is mentioned that the modeling results suggest an unchanged LGM volumetric census in both the western and eastern Atlantic.

Response #19: Although the modelling results of Gebbie (2014) suggest that the overall volumetric composition of northern and southern water masses in both western and eastern Atlantic are similarly unchanged between the LGM and modern-day, there is some zonal structure in the eastern basin that is not explicitly discussed by the author (please refer to page 11, line 6 of Gebbie, 2014). Indeed, the modelling results show a deep water layer which has a higher proportion of northern-sourced water occupying a depth range of ~2.5–3.5 km in the eastern basin of the North Atlantic (Supplementary Fig. 12 of Gebbie, 2014). Such a deep water layer is not evident in the western basin (main text Fig. 4 of Gebbie, 2014).

Also if the origin of the deep eastern Atlantic water is the Nordic Sea, then why is it not visible in site [1]?

Response #20: This point was previously raised by another reviewer and was discussed and resolved in a prior revision. The LGM deep Nordic water is proposed to occupy the depth range of 2–3.5 km in the eastern basin of the North Atlantic, between glacial northern-sourced intermediate waters and glacial Antarctic Bottom Water (AABW) (Yu et al., 2008). Site [1], the deepest core site from the North-East Atlantic (4.28 km), is suggested to reflect the influence of glacial AABW (see also line 271–275 in the main text).

o Figure 3b: I would suggest to remove cores [8] and [10] as they don't add much information but contribute in making the figure less clear.

Response #21: In order to be objective and transparent in our inclusion criteria explained in Fig. 3 caption, we have decided to keep cores [8] and [10] in Fig. 3b. Overall, the data from cores [8] and [10] do contribute to the coherent $^{231}\text{Pa}/^{230}\text{Th}$ signal in the West Atlantic over the last 25 kyr (Fig. 2a), and they are also used in the generation of the composite record (Supplementary Fig. 6b and 7). Removing cores [8] and [10] from Fig. 3b would complicate our aim of including all relevant data, which was highlighted as being very important during the review process.

Authors' references:

Broecker, W., Putnam, A.E., 2012. How did the hydrologic cycle respond to the two-phase mystery interval? *Quaternary Sci Rev* 57, 17-25.

Gebbie, G., 2014. How much did Glacial North Atlantic Water shoal? *Paleoceanography* 29, 190-209.

Yu, J.M., Elderfield, H., Piotrowski, A.M., 2008. Seawater carbonate ion-delta C-13 systematics and application to glacial-interglacial North Atlantic ocean circulation. *Earth Planet Sc Lett* 271, 209-220.